# BRCA2 associates with MCM10 to suppress PRIMPOL-mediated repriming and single-stranded gap formation after DNA damage

Zhihua Kang [1], Pan Fu [1,2,10], Allen L. Alcivar[1,8,10], Haiqing Fu [3], Christophe Redon[3], Tzeh Keong Foo [1], Yamei Zuo [1], Caiyong Ye[1,9], Ryan Baxley [4], Advaitha Madireddy [5], Remi Buisson [6,7], Anja-Katrin Bielinsky [4], Lee Zou [6], Zhiyuan Shen [1], Mirit I. Aladjem [3] & Bing Xia [1✉]

The BRCA2 tumor suppressor protects genome integrity by promoting homologous recombination-based repair of DNA breaks, stability of stalled DNA replication forks and DNA damage-induced cell cycle checkpoints. BRCA2 deficient cells display the radio-resistant DNA synthesis (RDS) phenotype, however the mechanism has remained elusive. Here we show that cells without BRCA2 are unable to sufficiently restrain DNA replication fork progression after DNA damage, and the underrestrained fork progression is due primarily to Primase-Polymerase (PRIMPOL)-mediated repriming of DNA synthesis downstream of lesions, leaving behind single-stranded DNA gaps. Moreover, we find that BRCA2 associates with the essential DNA replication factor MCM10 and this association suppresses PRIMPOL-mediated repriming and ssDNA gap formation, while having no impact on the stability of stalled replication forks. Our findings establish an important function for BRCA2, provide insights into replication fork control during the DNA damage response, and may have implications in tumor suppression and therapy response.

[1] Department of Radiation Oncology, Rutgers Cancer Institute of New Jersey, New Brunswick, NJ, USA. [2] Department of Clinical Microbiology Laboratory, Children's Hospital of Fudan University, Shanghai, China. [3] Developmental Therapeutics Group, Center for Cancer Research, National Cancer Institute, Bethesda, MD, USA. [4] Department of Biochemistry, Molecular Biology, and Biophysics, University of Minnesota, Minneapolis, MN, USA. [5] Department of Pediatric Hematology/Oncology, Rutgers Cancer Institute of New Jersey, New Brunswick, NJ, USA. [6] Massachusetts General Hospital Cancer Center and Harvard Medical School, Boston, MA, USA. [7] Department of Biological Chemistry, University of California Irvine, Irvine, CA, USA. [8] Present address: Bristol-Myers Squibb Company, Bloomsbury, NJ 08804, USA. [9] Present address: School of Radiation Medicine and Protection, Soochow University, Suzhou, China. [10] These authors contributed equally: Pan Fu, Allen L. Alcivar. ✉email: xiabi@cinj.rutgers.edu

BRCA2 is a major tumor suppressor with a critical role in genome integrity maintenance. Germline, monoallelic mutations in BRCA2 predispose to breast, ovarian and other cancers[1,2], whereas biallelic mutations in the gene cause a severe form of Fanconi anemia (FA-D1) characterized by development of solid tumors during the early childhood[3,4]. Early studies established a critical role of BRCA2 in the homologous recombination (HR)-based repair of DNA double strand breaks (DSBs)[5]. Then, BRCA2 mutant cells were found to display the radio-resistant DNA synthesis (RDS) phenotype[6], indicative of a possible intra-S-phase checkpoint defect. Later, it was shown that BRCA2 is required for the stability of stalled replication forks[7], and this has been a subject of intense studies in recent years[8–10]. Additionally, BRCA2 was also shown to facilitate fork restart after stalling[11]. However, whether BRCA2 regulates replication fork progression after DNA damage or replication stress remains unknown.

In mammals, DNA replication starts from tens of thousands of replication origins scattered throughout the genome. To ensure that DNA is replicated once and only once per cell cycle, the initiation of replication is a tightly controlled process that involves sequential and concerted actions of numerous factors[12]. Among these factors, the MCM2-7 helicase unwinds DNA, and MCM10, a unique member of the MCM family[13], interacts with and promotes the final activation of MCM2-7 helicase, leading to full DNA unwinding and origin firing[14]. In addition to origin activation, MCM10 has been reported to recruit DNA polymerase α (polα)[15,16], which catalyzes initial DNA synthesis on both leading and lagging strands at origins and the synthesis of Okazaki fragments on the lagging strands[17]. Once DNA replication commences, MCM10 "travels" with the replication fork to facilitate replication elongation[18]. More recently, MCM10 was also found to possess potent strand-annealing activity and to inhibit fork reversal[19].

In the present study, we investigated the mechanism of the RDS phenotype of BRCA2-deficient cells by using the DNA fiber assay[20] to analyze replication kinetics in cells with and without BRCA2. We found that BRCA2-deficient cells failed to sufficiently retrain replication fork progression after DNA damage, which likely underlies the RDS phenotype of the cells. Moreover, using affinity purification, we identified an association of BRCA2 with MCM10. In the data presented below, we focus on the regulation of replication fork progression by BRCA2 and the role of its association with MCM10 in the process.

## Results
### BRCA2 restrains replication fork progression after DNA damage.
To explore the role of BRCA2 in DNA replication fork progression, we examined the rate of fork elongation in U2OS cells depleted of BRCA2 (Fig. 1a) using the DNA fiber assay, in which replicating DNA was labeled with thymidine analogs CldU and IdU over a standard labeling period of 20 min and then visualized by staining with respective antibodies (Fig. 1b). Compared with cells treated with control siRNAs, BRCA2-depleted cells showed normal fork progression in unperturbed replication (Fig. 1c). At 6 h after 10 Gy of ionizing radiation (IR), control cells showed a substantial drop in replication tract length (Fig. 1c); while tract length was also reduced in BRCA2-depleted cells, it was significantly longer than that in control cells (Fig. 1c). We then conducted a time course experiment to assess the kinetics of fork progression in cells with and without BRCA2 after either 10 or 2 Gy of IR, and BRCA2-depleted cells showed significantly longer tract lengths at all the time points (1, 3, and 6 h) after IR (Fig. 1d, e).

Next, we investigated the impact of BRCA2 loss on fork progression after DNA damage induced by bleomycin (BLEO), a radiomimetic therapeutic DNA-damaging agent. Control and BRCA2-depleted U2OS cells were subjected to either a short (1 h) pulse with a high-concentration (10 μM) or a prolonged (6 h) and continuous treatment with a low concentration (1 μM) of the drug. Underrestrained fork progression was again seen in BRCA2-depleted cells under both conditions (Fig. 1f, g). Additionally, we also tested fork progression after continuous treatment with methylmethane sulfonate (MMS), an alkylating agent that causes base damage, camptothecin (CPT), a topoisomerase I inhibitor that produces covalent DNA-protein conjugates, and hydroxyurea (HU), which depletes cellular nucleotide pool leading to replication stress and ssDNA at the replication fork. Again, while all three drugs led to major reductions in fork speed in both control and BRCA2-depleted cells, the latter were found to have longer replication tracts after treatment with all three drugs (Fig. 1g). These findings demonstrate that the function of BRCA2 to suppress fork progression after DNA damage and/or replication stress is general and not specific to IR and radiomimetic agents.

To further confirm the role of BRCA2 in restraining fork progression after DNA damage, we analyzed fork progression in VC8 cells, a Chinese hamster ovary (CHO) cell line with biallelic BRCA2 mutations[6] and the same cells reconstituted with a human BRCA2 cDNA[7] (Fig. 1h). While re-expression of BRCA2 in VC8 cells did not produce any effect on fork velocity during unperturbed replication, it significantly reduced replication tract length after either IR or BLEO treatment (Fig. 1i, j). Taken together, these results demonstrate that BRCA2 restrains replication fork progression after DNA damage.

### BRCA2 suppresses primase-polymerase (PRIMPOL)-mediated repriming after DNA damage.
Underrestrained fork progression after DNA damage may result from several distinct mechanisms. First, we considered fork reversal, as higher overall fork speed can result from defective fork reversal[21]. Although BRCA2 protects (stalled and) reversed forks, it is not required for fork reversal per se[8–10]. Several fork remodeling factors, including HLTF[22], SMARCAL1[23], and ZRANB3[21] are required for the formation of reversed forks. Therefore, we used siRNAs to downregulate these fork remodeling factors (Fig. 2a) and measured the impact on fork progression in the presence and absence of BRCA2. While loss of none of these factors showed any effect on fork progression in unirradiated cells (Fig. 2b), depletion of each of them led to higher overall fork velocity in irradiated cells (Fig. 2c), indicating that fork reversal indeed contributed to fork slowdown upon DNA damage. Co-depletion of each of these factors with BRCA2 caused further increased fork velocity, showing mostly an additive effect (Fig. 2d). These results confirm that fork reversal operates normally in the absence of BRCA2 and, more importantly, they indicate that the increased fork progression in BRCA2-depleted cells results from a different mechanism.

Recent studies have established primase-polymerase (PRIMPOL)[24], which possesses an activity to reprime DNA synthesis downstream of lesions, as a key regulator of replication fork progression after DNA damage[25]. A recent study also demonstrates that PRIMPOL is induced after cisplatin treatment and promotes lesion skipping and prevents fork reversal and degradation in BRCA1-deficient cells[26]. Therefore, we asked if PRIMPOL is responsible for the underrestrained fork progression observed in BRCA2-deficient cells after IR, as repriming and lesion skipping would prevent fork stalling, thereby increasing fork progression. siRNA-mediated depletion of PRIMPOL alone

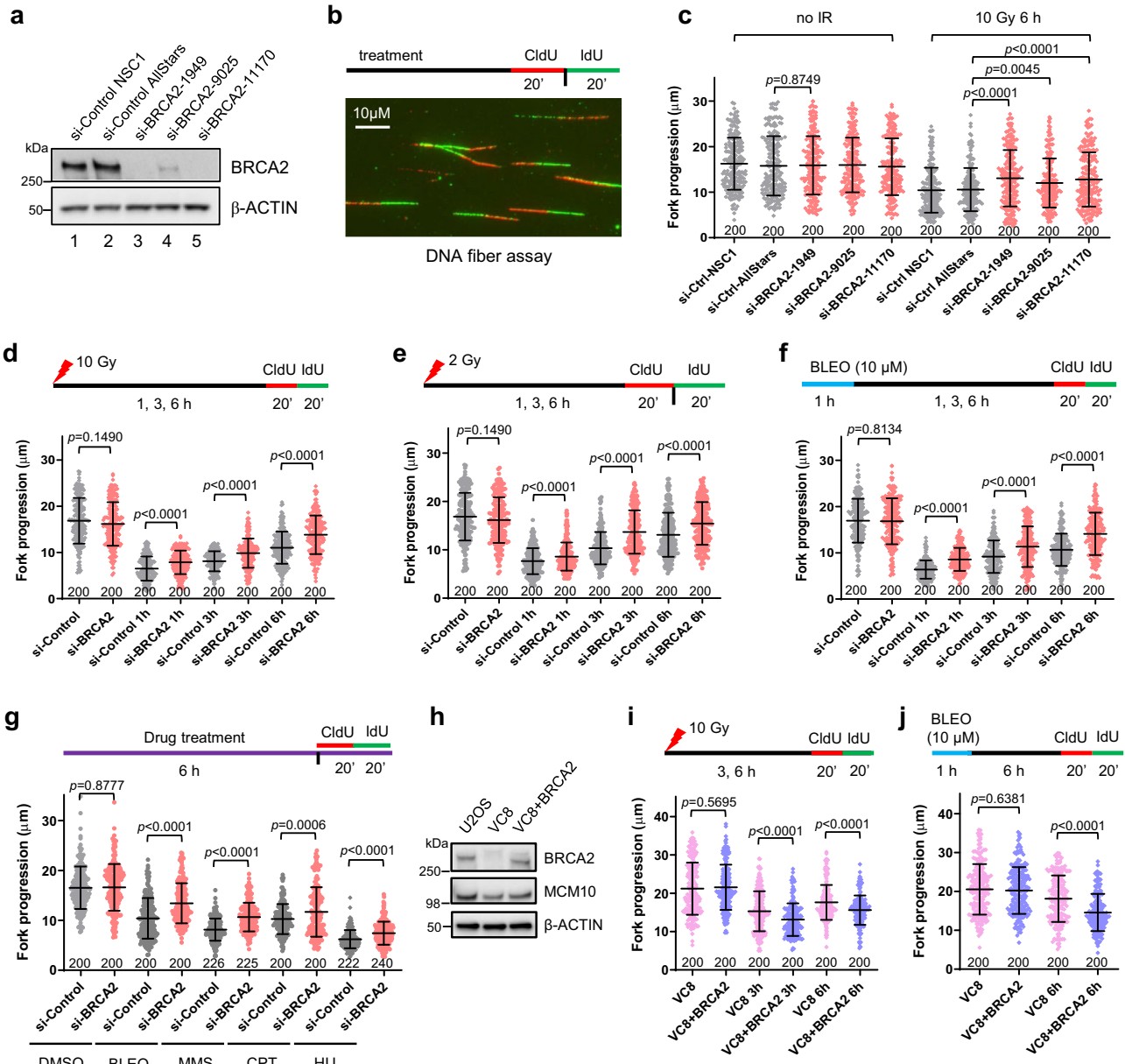

**Fig. 1 BRCA2 restrains replication fork progression after DNA damage. a** Western blots showing levels of BRCA2 in U2OS cells treated with 2 different control and 3 different BRCA2 siRNAs. Similar results were obtained in $n = 3$ independent experiments. **b** Labeling scheme and a representative image of the DNA fiber assay. **c** Lengths of IdU-labeled replication tracts in U2OS cells treated with control and BRCA2 siRNA before and 6 h after 10 Gy of IR. **d–f** Replication tract lengths in U2OS cells treated with control (NSC1) and BRCA2 (1949) siRNAs before and 1, 3, and 6 h after treatment with 10 Gy of IR (**d**), 2 Gy of IR (**e**), or 10 μM of bleomycin (BLEO) (**f**). **g** Replication tract lengths in control and BRCA2-depleted U2OS cells after 6 h of continuous treatment with DMSO, BLEO (1 μM), MMS (50 μM), CPT (1 μM), and HU (50 μM). **h** Levels of BRCA2 and MCM10 in U2OS, VC8, and VC8 cells reconstituted with a human BRCA2 cDNA. **i, j** Replication tract lengths in VC8 and VC8 + BRCA2 cells before and 3 and 6 h after 10 Gy of IR (**i**) or 6 h after BLEO treatment (**j**). Data in (**c–g**) and (**i, j**) are presented as mean ± standard deviation (s.d.), with the number of dots shown below each column. Similar results were obtained from $n = 2$ biologically independent experiments. $P$ values are calculated using two-tailed unpaired Student's $t$ test. Source data are provided as a Source Data file.

showed little to no effect on fork velocity either before or after IR (Fig. 2e, f); however, when PRIMPOL was co-depleted with BRCA2, the increased fork progression elicited by BRCA2 loss was completely reversed (Fig. 2f). Transient re-expression of wild-type (WT) PRIMPOL in the co-depleted cells restored the increased fork progression after IR, whereas the CH (primase dead) and AxA (catalytic dead) mutants[27] both failed to do so (Supplementary Fig. 1), implying that both primase and polymerase activities of PRIMPOL are required for the

underrestrained replication fork progression in BRCA2-depleted cells after DNA damage.

Repriming of DNA synthesis by PRIMPOL generates ssDNA gaps on replicated DNA as a result of lesion bypass[25]. We therefore carried out a modified DNA fiber assay, in which cells were treated with the ssDNA-specific S1 endonuclease after being labeled with the thymidine analogs[28]. In this assay, any shortening of labeled DNA tracts after S1 treatment would be evidence of the presence of ssDNA gaps. Indeed, significant shortening of

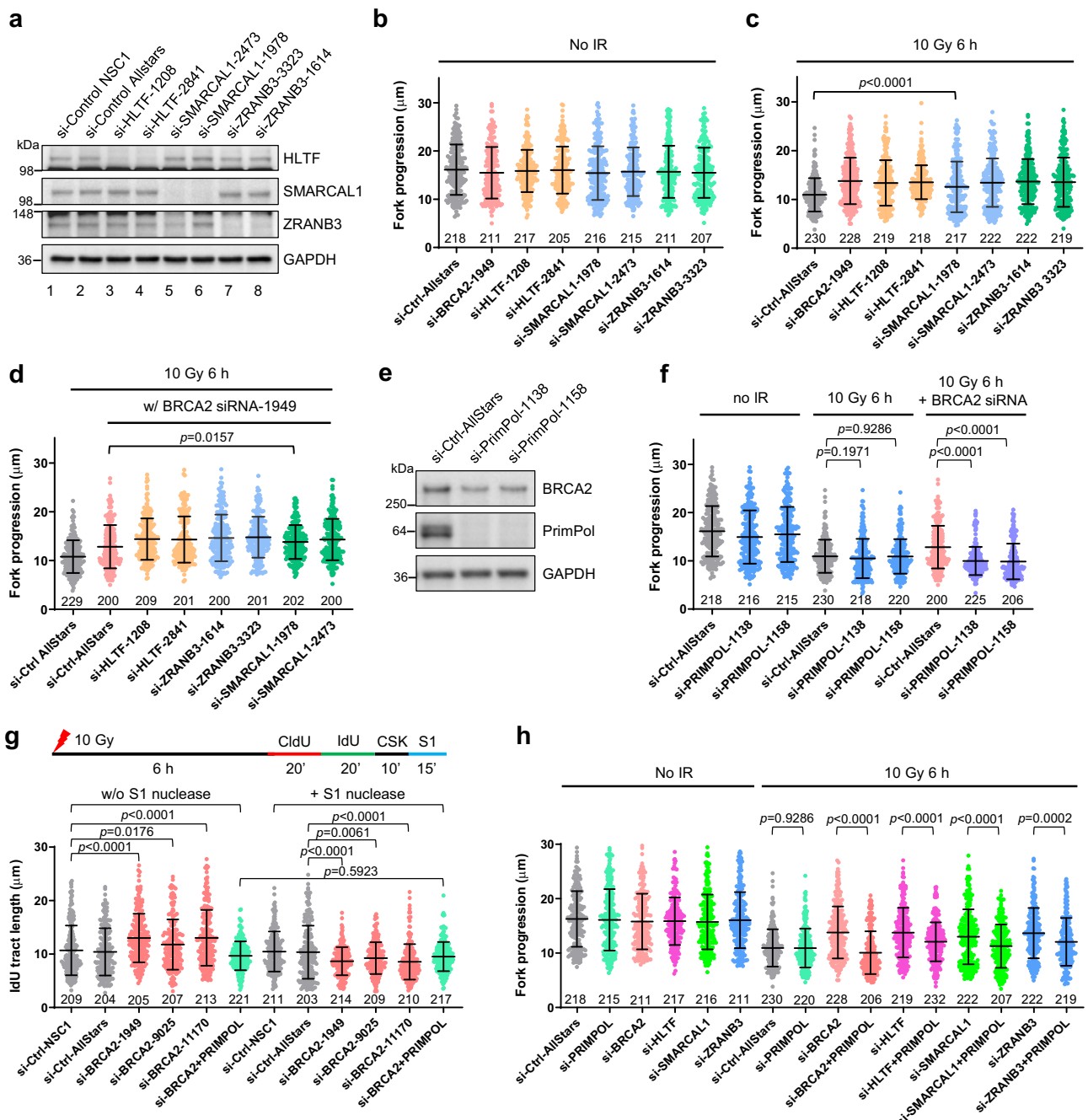

**Fig. 2 BRCA2 suppresses PRIMPOL-mediated repriming and ssDNA gap formation after DNA damage. a** Western blots showing siRNA-mediated depletion of HLTF, SMARCAL1, and ZRANB3 in U2OS cells. Two siRNAs for each gene were used. Similar results were obtained in $n = 3$ independent experiments. **b, c** Replication tract lengths in U2OS cells depleted of BRCA2, HLTF, SMARCAL1, or ZRANB3 either before (**b**) or 6 h after (**c**) 10 Gy of IR. **d** Replication tract lengths in U2OS cells co-depleted of BRCA2 with HLTF, SMARCAL1, or ZRANB3 at 6 h after 10 Gy of IR. **e** Western blots showing the levels of PRIMPOL depletion with two different siRNAs. Similar results were obtained in $n = 3$ independent experiments. **f** Replication tract lengths in U2OS cells depleted of PRIMPOL alone or co-depleted of PRIMPOL and BRCA2 before and 6 h after 10 Gy of IR. **g** Effect of S1 nuclease treatment on replication tract lengths in control and BRCA2-depleted U2OS cells at 6 h after 10 Gy of IR. **h** Replication tract lengths in U2OS cells depleted of BRCA2, HLTF, SMARCAL1, or ZRANB3 alone or in combination with PRIMPOL at 6 h after 10 Gy of IR. Data in (**b–d**) and (**f–h**) are presented as mean ± s.d., with the number of dots shown below each column. *P* values are calculated using two-tailed unpaired Student's *t* test. Source data are provided as a Source Data file.

replication tracts was seen in irradiated BRCA2-depleted cells after S1 nuclease treatment, whereas in cells transfected with control siRNAs, S1 treatment did not cause any change (Fig. 2g). Moreover, in cells depleted of both BRCA2 and PRIMPOL, S1 nuclease treatment also did not produce any difference (Fig. 2g). These results clearly demonstrate ssDNA gap formation in

BRCA2-depleted cells after IR and further indicate that PRIMPOL-mediated repriming underlies the underrestrained fork progression in BRCA2-deficient cells after DNA damage.

Finally, we tested whether PRIMPOL-mediated repriming was operative in cells depleted of HLTF, SMARCAL1, and ZRANB3 after IR. We found that co-depletion of PRIMPOL with each of

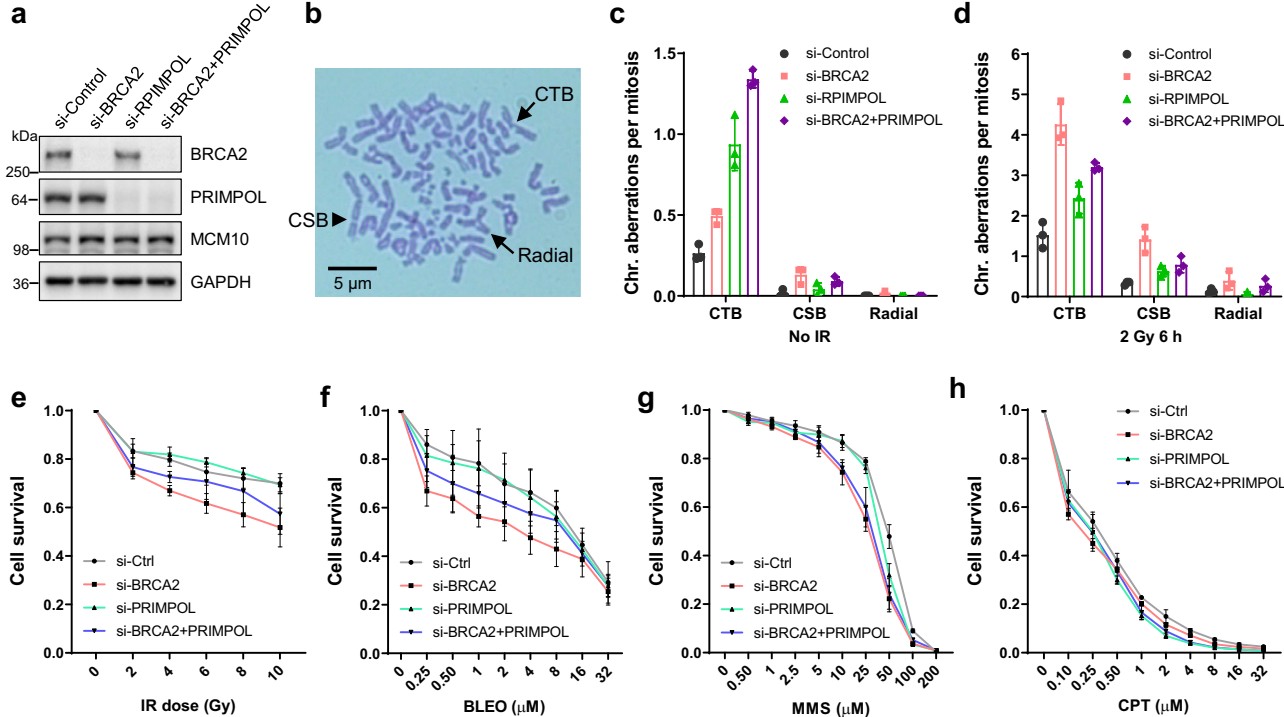

**Fig. 3 Impact of PRIMPOL loss on genome stability and cell viability after DNA damage in BRCA2-proficient and -deficient cells. a** Western blots showing levels of siRNA-mediated depletion of BRCA2, PRIMPOL, or BRCA2 + PRIMPOL in U2OS cells. Similar results were obtained in $n = 3$ independent experiments. **b** A representative image of a mitotic spread after Giemsa staining showing different forms of abnormal chromosomes. CTB, chromatid break; CSB, chromosomal break; Radial, radial chromosome. **c, d** Quantification of chromosomal aberrations in U2OS cells treated with control, BRCA2, PRIMPOL, or BRCA2 + PRIMPOL siRNAs without IR (**c**) or 6 h after 2 Gy of IR (**d**). Data are presented as mean ± s.d. from $n = 3$ independent experiments. **e–h** Sensitivity of the above siRNA-treated cells to IR (**e**), BLEO (**f**), MMS (**g**), and CPT (**h**). Cell viability was determined 72 h after treatment. Results are presented as mean ± s.d. from $n = 3$ independent experiments, each conducted in technical triplicates. Source data are provided as a Source Data file.

the three factors significantly reversed the increased fork progression after IR (Fig. 2h). This suggests that repriming by PRIMPOL contributes to increased fork progression after DNA damage in the absence of fork reversal; in other words, fork reversal precludes PRIMPOL from being recruited and/or acting at damaged forks.

**Impact of PRIMPOL loss on genome stability and cell viability after DNA damage in the presence and absence of BRCA2.** To determine the role of PRIMPOL and repriming in genome stability in BRCA2-deficient cells, we depleted the two proteins, alone and in combination (Fig. 3a), in U2OS cells and analyzed mitotic spreads (Fig. 3b) for genome instability. In the absence of external DNA damage, cells depleted of either PRIMPOL or BRCA2 both showed increased genome instability, primarily in the form of chromatid breaks (CTBs), as compared with cells treated with a control siRNA (Fig. 3c). A small number of chromosomal breaks (CSBs) were also detected in PRIMPOL- and BRCA2-depleted cells, whereas radial chromosomes were practically not observed. Following IR (2 Gy, 6 h), overall genome instability was elevated in all cells, with PRIMPOL- and BRCA2-depleted cells still showing higher levels of genome instability than control cells (Fig. 3d). Interestingly, PRIMPOL-depleted cells harbored more CTBs than did BRCA2-depleted cells prior to IR but less after IR. This suggests that in unperturbed cells, ssDNA gaps generated by PRIMPOL leads to more DSBs than does the inability of BRCA2-depleted cells to repair DSBs due to their HR deficiency, whereas after radiation-induced DNA damage, loss of HR-mediated DSB repair has a greater impact on the amount of unrepaired DNA breaks in the cell. Moreover, combined loss of PRIMPOL and BRCA2 led to further increased

CTBs compared with individual loss of either protein before radiation, but reduced CTBs compared with BRCA2 loss after radiation. This suggests that the effects of loss of the two proteins may be additive in unperturbed cells, whereas after radiation, loss of PRIMPOL and the resulting loss of repriming and ssDNA formation may prevent DNA breakage (at regions of ssDNA), thereby reducing CTBs. In irradiated cells, the levels of CSBs were still much lower than those of CTBs but appeared to follow the same trend as CTBs. Radiation also led to the formation of a small number of radial chromosomes, with BRCA2 loss causing an increase in this form of chromosomal abnormality while PRIMPOL depletion having practically no effect (Fig. 3d).

Finally, we analyzed the sensitivity of the above cells to IR, BLEO, MMS, and CPT (Fig. 3d–g). As expected, loss of BRCA2 led to increased sensitivity to all four DNA-damaging agents, while depletion of PRIMPOL showed no effect, consistent with previous reports[29,30]. Notably, co-depletion of PRIMPOL with BRCA2 led to a partial rescue of the hypersensitivity of BRCA2-depleted cells to radiation and bleomycin but not MMS and CPT, suggesting that PRIMPOL or ssDNA gaps resulting from its repriming activity contributes to cellular sensitivity after certain types of DNA damage, which remains to be fully defined.

**Identification of MCM10 as a BRCA2 binding protein.** We have previously used tandem affinity purification (TAP) coupled with mass spectrometry (MS) to identify new association partners of PALB2, a major BRCA2 binding protein that is required for its chromatin association, DNA damage-induced foci formation, and HR activity[31]. The efforts led to the identification of BRCA1 and KEAP1 as PALB2 binding proteins[32,33]. However, in these previous experiments, individual bands or gel sections were analyzed

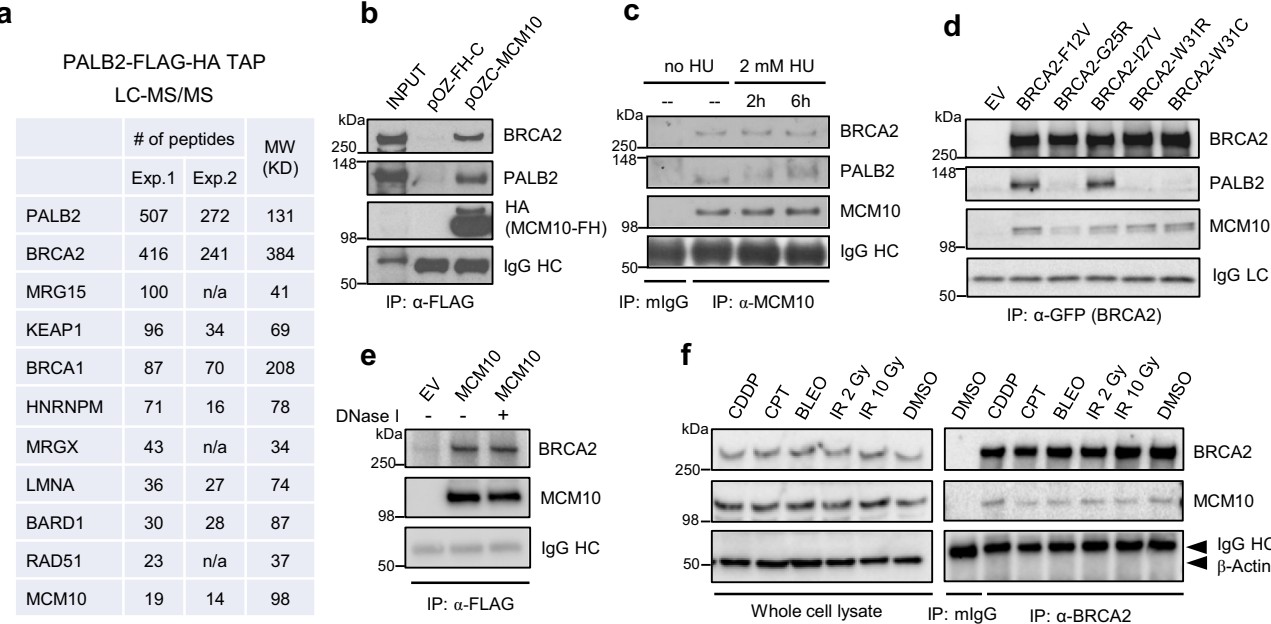

**Fig. 4 Identification of MCM10 as a BRCA2-ineracting protein. a** Numbers of tryptic peptides obtained from two independent PALB2 tandem affinity purification (TAP)-LC-MS/MS experiments. **b** Co-IP of endogenous PALB2 and BRCA2 with ectopic MCM10. FLAG-HA-tagged MCM10 was transiently overexpressed in 293T cells and IPed with anti-FLAG. **c** Co-IP of endogenous MCM10 and endogenous PALB2 and BRCA2 in unperturbed and HU-treated U2OS cells. **d** Co-IP of GFP-tagged BRCA2 variants and FLAG-HA-tagged MCM10 overexpressed in 293T cells. **e** Association of endogenous BRCA2 with ectopic FLAG-HA-tagged MCM10. The procedure was performed as in (**b**) except that the final IP material was treated with DNase I on the beads and then washed before being subjected to western blotting. **f** Co-IP of BRCA2 and MCM10 in S-phase-synchronized U2OS cells at 3 h after treatment with DMSO, 10 Gy of IR, 2 Gy of IR, BLEO (1 μM), CPT (1 μM), or cisplatin (CDDP, 1 μM). Cells were synchronized in the S phase by a single thymidine block for 24 h and released for 2 h prior to the treatments. For (**b–f**), similar results were obtained in n = 3 independent experiments. Source data are provided as a Source Data file.

by MS, leading to incomplete identification of components of the complex. To further identify new components of the PALB2 complex, we reconducted the purification and subjected the entire content of the complex to MS analysis. As shown in Fig. 4a, in addition to known interactors such as BRCA2, BRCA1 (and its close partner BARD1), KEAP1, RAD51, MRG15, and MRGX[34], our new TAP-MS analyses identified HNRNPM, LMNA, and MCM10 as candidate new components of the PALB2 complex. We decided to focus on MCM10, as it is an essential DNA replication factor[13].

To confirm the association between MCM10 and PALB2, we cloned the human MCM10 cDNA into a FLAG-HA double tagging vector, transiently expressed and immunoprecipitated (IPed) the protein, and tested the presence of endogenous PALB2 and BRCA2 in the precipitates. Indeed, both PALB2 and BRCA2 were readily detectable (Fig. 4b). We then IPed endogenous MCM10 and again detected both PALB2 and BRCA2 in the precipitates (Fig. 4c). Therefore, MCM10 is a bona fide component of the PALB2/BRCA2 complex. HU treatment did not produce any significant effect on the association between MCM10 and PALB2/BRCA2, although it triggered PALB2 phosphorylation.

Given the known role of BRCA2 in DNA replication and the relatively large amount of BRCA2 associated with PALB2, we asked if MCM10 in fact interacts with BRCA2. To this end, we tested a panel of patient-derived BRCA2 variants in its PALB2-binding motif[31] for their ability to associate with MCM10. As shown in Fig. 4e, MCM10 was found to co-IP with not only PALB2 binding neutral variants (F12V and I27V) but also defective variants (G25R, W31R, and W31C), indicating that it indeed can associate with BRCA2 independently of PALB2. To rule out the possibility that the BRCA2–MCM10 association may

be mediated by DNA in the cell lysate, we treated the IPed material on anti-FLAG beads with DNase I after the IP was completed, and the association remained intact (Fig. 4e). To determine the impact of DNA damage on the MCM10–BRCA2 association during DNA replication, we synchronized cells in the S phase, exposed them to different genotoxic agents, including IR, BLEO, CPT and cisplatin (a crosslinker), and then analyzed the association by BRCA2 IP. The results showed that the association remained largely unchanged after DNA damage induced by these agents (Fig. 4f).

**The N-terminal coiled-coil motif of MCM10 is required for its BRCA2 association**. To determine the structural element of MCM10 that is responsible for its association with BRCA2, we first generated a series of six deletions spanning the entire MCM10 coding sequence in a myc-MCM10-GFP vector (Supplementary Fig. 2a) and performed transient overexpression and IP-western. Surprisingly, BRCA2 was found to co-IP with all six deletion proteins under the conditions used (Supplementary Fig. 2b). This was likely an artifact as the proteins were expressed at very high levels. As we had been studying the BRCA1–PALB2 association mediated by their respective coiled-coil (CC) motifs[35], we then took a candidate approach and focused on the N-terminus, which contains a conserved CC motif (Fig. 5a). Deletion of this motif abrogated the co-IP of BRCA2 with MCM10, indicating that it is required for their association (Fig. 5b). It has been reported that the CC motif of *Xenopus* MCM10 mediates its dimer/oligomerization, and mutations of conserved hydrophobic residues in the motif abrogate this self-association[36]. We therefore generated the corresponding mutations in the human MCM10 and tested their effect on BRCA2 association. Indeed, the 2D (L114D/L118D) and 4A (L114A/L118A/M125A/L128A)

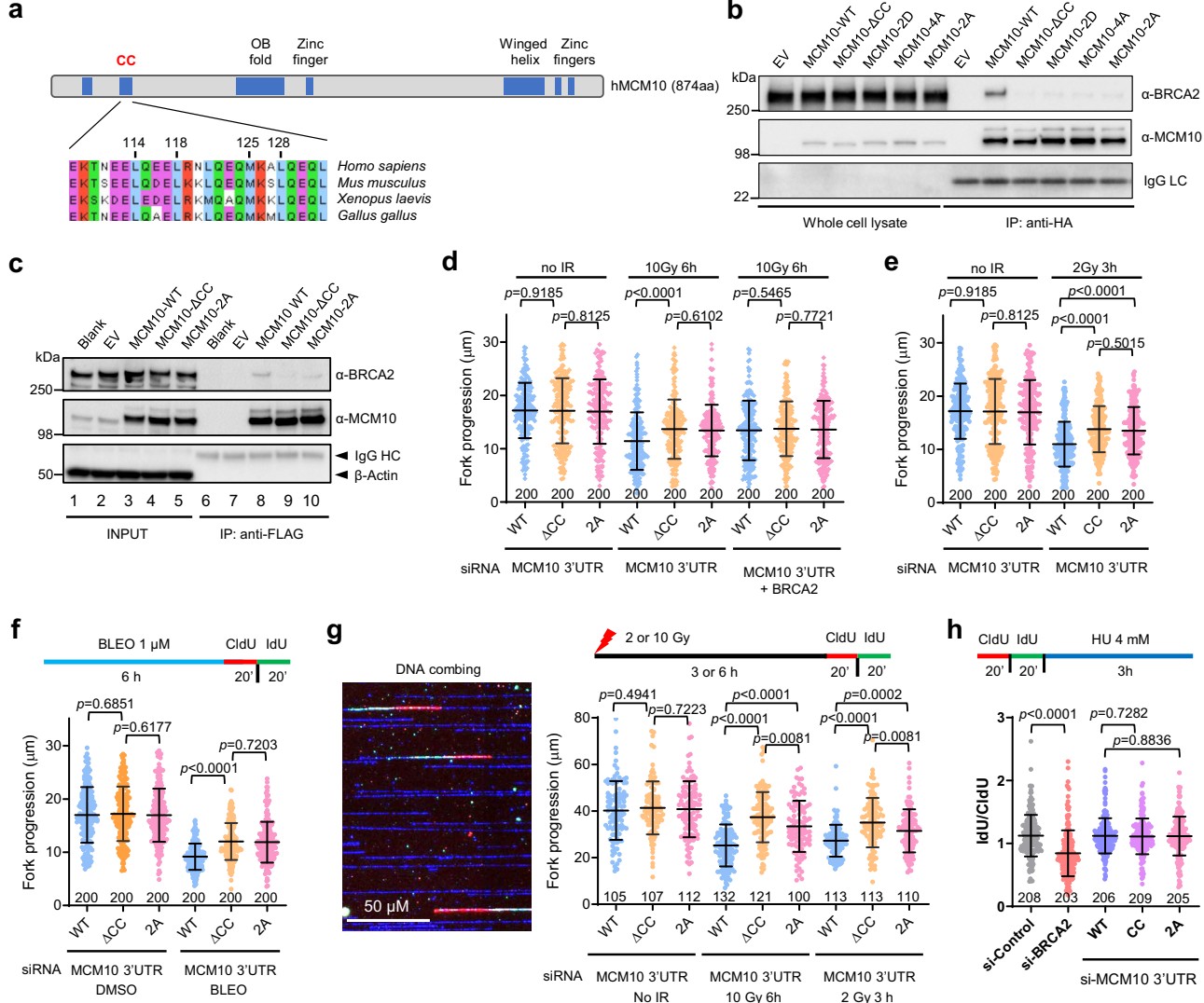

**Fig. 5 The BRCA2–MCM10 association restrains replication fork progression after DNA damage. a** Schematic diagram of MCM10 domain structure and amino acid sequence alignment of the MCM10 N-terminal coiled-coil (CC) motif. **b** Co-IP of endogenous BRCA2 with FLAG-HA-tagged MCM10 proteins transiently overexpressed in 293T cells. **c** Expression levels and association between exogenous MCM10 proteins and endogenous BRCA2 in U2OS cell lines stably expressing FLAG-HA-tagged WT and CC mutant MCM10 proteins. For (**b**) and (**c**), similar results were obtained in $n = 3$ independent experiments. **d–f** Replication tract lengths in U2OS cells selectively expressing exogenous WT or mutant MCM10 proteins before or 6 h after 10 Gy of IR (**d**), 3 h after 2 Gy of IR (**e**), or after 6 h of continuous treatment with DMSO (vehicle) or 1 μM BLEO (**f**). **g** Replication tract lengths in the above cells before and after IR as determined by the DNA combing assay. A representative combing image is shown on the left. **h** Stability of stalled replication forks in the above cells. Assay scheme is shown on the top, and IdU/CldU tract length ratio after HU treatment is shown below. Data in (**d–h**) are presented as mean ± s.d., with the number of dots shown below each column. *P* values are determined two-tailed unpaired Student's *t* test. Similar results were obtained from $n = 2$ independent experiments. Source data are provided as a Source Data file.

mutations both led to loss of BRCA2 binding (Fig. 5b). Thus, it appears that the same residues are involved in both MCM10–MCM10 and BRCA2–MCM10 associations. Additionally, we also tested a 2A (L114A/L118A) mutation, which was shown to have no impact on MCM10 self-association[36]. Interestingly, this mutation also abrogated the BRCA2 association with MCM10 (Fig. 5b).

**BRCA2–MCM10 association restrains replication fork progression after DNA damage.** To determine the role of the BRCA2–MCM10 association in DNA replication, we constructed stable U2OS cell lines expressing MCM10-WT, MCM10-ΔCC, or MCM10-2A (Fig. 5c). In this setting, MCM10-ΔCC remained largely unable to associate with BRCA2, whereas the 2A mutant showed significant residual association. We then depleted the

endogenous MCM10 in the stable cell lines using a pool of two siRNAs against the 3′-UTR of its mRNA (Supplementary Fig. 3) and measured replication fork progression supported by the exogenous proteins. No significant difference was detected among the cells during unperturbed replication (Fig. 5d); after DNA damage elicited by either IR or BLEO, however, cells selectively expressing either mutant MCM10 protein showed underrestrained fork progression (Fig. 5d–f).

To confirm the above findings, we conducted the DNA combing analysis, in which DNA molecules would be stretched more fully than in the DNA fiber assay, under the same conditions as above. As shown in Fig. 5g, replication tracts determined by DNA combing were 2–2.5 times longer than those determined by the fiber assay, and the difference between cells expressing WT and mutant MCM10 proteins were even more

profound. Moreover, while replication tracts in cells expressing MCM10-2A were evidently longer than those in cells expressing the WT protein, they were significantly shorter than those in cells expressing the ΔCC mutant, a phenotype consistent with its partial loss of BRCA2 binding capacity. Collectively, the above results suggest that BRCA2 inhibits fork progression after DNA damage through its association with MCM10.

To rule out the possibility that the underrestrained fork progression in cells expressing the mutant MCM10 proteins was due to compromised MCM10 dimer/oligomerization or other functions of the CC motif, we also co-depleted BRCA2 with the endogenous MCM10 in the above stable cells and analyzed fork progression using DNA fiber assay. Under this condition, the differences between cells expressing WT and the mutant MCM10 proteins were no longer observed (Fig. 5d). Moreover, depletion of endogenous BRCA2 did not further accelerate fork progression in cells stably expressing the mutant MCM10 proteins. Thus, it is the association of BRCA2 with the CC motif of MCM10 that restrains fork progression after DNA damage.

As BRCA2 is required for the stability of stalled replication forks, we next asked whether its association with MCM10 contributes to its fork stabilizing function. We labeled the above cells selectively expressing exogenous WT and mutant MCM10 proteins, treated them with HU, and then determined the lengths of CldU- and IdU-labeled tracts (Fig. 5h). Blank U2OS cells depleted of BRCA2 were used as positive control. Loss of BRCA2 led to substantial shortening of newly synthesized DNA tracts after HU treatment, indicative of a degradation of stalled forks, whereas loss of BRCA2–MCM10 association had no impact (Fig. 5h). As such, the association between BRCA2 and MCM10 is required for the cell to restrain replication fork progression after DNA damage but is not involved in fork stabilization upon stalling.

**BRCA2–MCM10 association prevents PRIMPOL-mediated repriming after DNA damage**. Finally, we asked if the function of BRCA2 to suppress PRIMPOL-mediated repriming depends on its association with MCM10. We depleted PRIMPOL together with the endogenous MCM10 in the above stable cell lines expressing WT or MCM10 CC mutants (Supplementary Fig. 3) and measured the lengths of replication tracts. Depletion of PRIMPOL alone had no effect on fork progression in unperturbed cells but evidently reversed the increase in cells expressing MCM10 mutants after IR (Fig. 6a). Moreover, S1 nuclease treatment also reduced tract lengths in cells expressing the mutant MCM10 proteins, without any impact on cells expressing the WT protein (Fig. 6b). Taken together, these results indicate that BRCA2 functions through its association with MCM10 to restrain fork progression after DNA damage by suppressing PRIMPOL-mediated repriming and lesion skipping.

## Discussion

In this study, we found that BRCA2-deficient cells underwent underrestrained replication fork progression after being treated with diverse types of DNA-damaging agents. Subsequently, we found that the increased fork progression in BRCA2-deficient cells was due to PRIMPOL-mediated replication repriming, leaving behind ssDNA gaps in the newly synthesized strand. Furthermore, we identified an association of BRCA2 with MCM10 mediated by the N-terminal CC motif of the latter. Loss of this association led to underrestrained fork progression after DNA damage, as cells in which the endogenous MCM10 was replaced by exogenous MCM10 lacking the CC motif behaved in the same fashion as did cells depleted of BRCA2, and depletion of BRCA2 eliminated the difference between cells expressing WT and the mutant MCM10 without causing further increase in fork progression in cells expressing the mutant proteins. Additionally,

underrestrained fork progression in cells selectively expressing the mutant MCM10 proteins was also found to be due to PRIMPOL-mediated repriming. Based on these findings, we propose a model wherein BRCA2 is recruited to replication forks by MCM10 and, upon stalling of DNA polymerase ε at DNA lesions, inhibits PRIMPOL-mediated repriming, thereby restraining fork progression and preventing ssDNA gap formation (Fig. 6c).

How does BRCA2 inhibit repriming by PRIMPOL? One possible scenario is that it may physically and directly block PRIMPOL recruitment after its own recruitment by MCM10, given its very large size (~384 KD). As it has been shown that PRIMPOL is recruited by replication protein A (RPA) to ssDNA[37] and that BRCA2/DSS1 can displace RPA from ssDNA[38], it is also possible that BRCA2 may indirectly prevent PRIMPOL recruitment to damaged forks by removing RPA from the ssDNA arising from the uncoupling of DNA polymerase ε and the MCM2-7 helicase complex upon encounter of DNA lesions. In supporting these two possible scenarios, the amount of chromatin-associated PRIMPOL was larger in cells selectively expressing the MCM10 mutants than in cells expressing the WT protein after DNA damage (Supplementary Fig. 4), although this remains to be substantiated by more precise methodologies such as iPOND[39]. Besides, the possibility that BRCA2 or the BRCA2/MCM10 complex may directly or indirectly inhibit PRIMPOL's enzymatic activity cannot be ruled out.

The exact lesions that are bypassed by PRIMPOL in the absence of BRCA2 or BRCA2–MCM10 association also remain to be further defined. IR can cause a variety of DNA damage including base damage, single strand breaks (SSBs), DSBs, and DNA–DNA or DNA–protein crosslinks[40], and bleomycin, much like IR, can generate abasic site, SSBs, and DSBs[41]. While it is unlikely for PRIMPOL to reprime across DSBs, any form of DNA damage that leads to ssDNA regions behind the replication fork could potentially trigger RPA-mediated PRIMPOL recruitment and repriming. Indeed, we found that BRCA2-depleted cells also showed underrestrained fork progression after treatment with MMS, CPT, and HU (Fig. 1g). These findings indicate that PRIMPOL can catalyze repriming across a variety of DNA lesions as a general mechanism for BRCA2-deficient cells to bypass DNA lesions other than DSBs. Careful analyses of repriming efficiency following treatment of BRCA2-deficient cells with different DNA agents at different doses are required to further define the preferred lesions for PRIMPOL action and the relative efficiency of repriming across different types of lesions.

Another key question is whether this newly discovered BRCA2 function is shared by RAD51 and BRCA1, its key partners in the HR pathway that have also been shown to play important roles in fork reversal and/or stability[42]. In this regard, a previous study has shown that loss of RAD51 leads to PRIMPOL-dependent unrestrained replication progression and ssDNA gaps after UV irradiation[43]. Therefore, we tested the impact of RAD51 loss on fork progression after IR. Indeed, RAD51 depletion led to longer replication tracts after IR; however, the underrestrained fork progression caused by RAD51 loss was barely reduced by co-depletion of PRIMPOL (Supplementary Fig. 5a, b). This indicates a limited involvement of PRIMPOL in the process and further suggests that the increased fork progression in RAD51-depleted cells may be mainly due to lack of fork reversal. A recent study has also shown PRIMPOL-mediated repriming in BRCA1-deficient cells, where PRIMPOL was found to be transcriptionally induced after cisplatin treatment and then reprime DNA synthesis[26]. This study was focused on PRIMPOL-mediated repriming to preclude fork reversal and degradation of reversed forks, while in the current study we present a mechanism that is not only specific to BRCA2 but also independent of fork reversal. In fact, we found that depletion of BRCA1 in U2OS cells led to

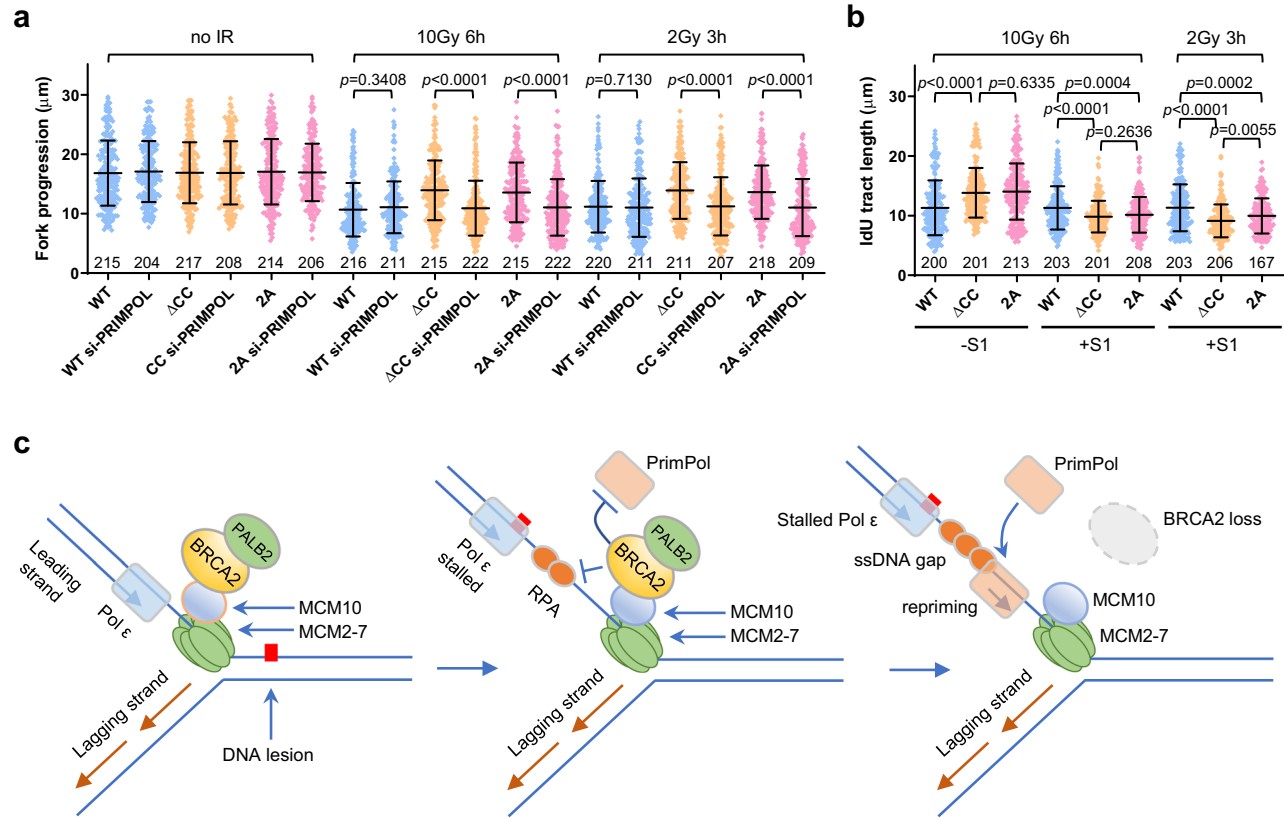

**Fig. 6 The BRCA2–MCM10 association suppresses PRIMPOL-mediated repriming and ssDNA gap formation after DNA damage. a** Effect of PRIMPOL depletion on replication fork progression in stable U2OS cells selectively expressing exogenous WT vs ΔCC mutant MCM10 proteins before and after IR. The endogenous MCM10 was depleted together with PRIMPOL with siRNAs targeting the 3′-UTR of its mRNA. **b** Effect of S1 nuclease treatment on detected replication tract length in the above cells after IR. Data in (**a**) and (**b**) are presented as mean ± s.d., with the number of dots shown below each column. P values are calculated using two-tailed unpaired Student's t test. Similar results were obtained in n = 2 independent experiments. **c** A proposed model for the role of BRCA2 and the BRCA2–MCM10 association in replication fork progression after DNA damage. BRCA2 is recruited to the replication fork via its association with MCM10. When the fork encounters a lesion, polymerase ε stalls, whereas the MCM2-7 helicase complex continues to unwind downstream DNA, leading to excessive ssDNA formation. RPA binds to the ssDNA and may recruit PRIMPOL; however, PRIMPOL recruitment is prevented by BRCA2, which can displace RPA from ssDNA. When BRCA2 is lost or the BRCA2–MCM10 association is compromised, increased and/or persistent presence of RPA on ssDNA leads to PRIMPOL recruitment, hence repriming of DNA synthesis downstream of the lesion and formation of ssDNA gaps. See text for further details.

substantially reduced fork progression even without exogenous DNA damage and that fork speed in BRCA1-depleted cells was further reduced and remained slower than control after IR (Supplementary Fig. 5c, d). Taken together, available evidence suggests that BRCA1, BRCA2, and RAD51 each has distinct roles with respect to PRIMPOL-dependent lesion bypass, and the role of BRCA2 to inhibit PRIMPOL-mediated repriming is likely independent of its HR function.

Finally, as noted before, this study began as an effort to elucidate the mechanism of the RDS phenotype and possible intra-S-phase checkpoint defect of BRCA2-deficient cells. To address RDS, we initially depleted BRCA2 and used BrdU incorporation to semi-quantitatively measure DNA synthesis before and after IR. BRCA2-depleted cells showed similar levels DNA synthesis during normal growth but indeed incorporated more BrdU than control cells after IR (Supplementary Fig. 6a–c). Consistent with its being a critical replication factor acting in both replication initiation and elongation, depletion of MCM10 led to reduced DNA synthesis, as measured by BrdU incorporation (Supplementary Fig. 6a–c), and slower replication fork progression, as determined by the DNA fiber assay (Supplementary Fig. 6d), both before and after IR. Moreover, cells selectively expressing MCM10-ΔCC also showed increased BrdU incorporation than

did cells expressing the WT protein (Supplementary Fig. 6e). Separately, we conducted DNA fiber analysis using a modified labeling scheme, in which we first labeled BRCA2-depleted cells with CldU for 20 min and then with IdU for 40 min in the presence of bleomycin. Under this condition, BRCA2-depleted cells as expected showed PRIMPOL-dependent increase in fork progression and ssDNA gap formation (Supplementary Fig. 7a), while no increase in origin firing was observed during bleomycin treatment (Supplementary Fig. 7b). As such, underrestrained fork progression, rather than increased origin firing, is the primary contributor to the RDS phenotype of BRCA2-deficient cells.

## Methods

**Cell lines and culture**. U2OS and HEK293T cells were purchased from ATCC and cultured in Dulbecco's modified Eagle's medium (DMEM) supplemented with 10% heat-inactivated fetal bovine serum (FBS) and penicillin-streptomycin (Pen-Strep). VC8 and VC8 + BRCA2 cells were provided by Dr. Maria Jasin (Memorial Sloan Kettering Cancer Center) and grown in DMEM/F12 supplemented with 10% heat-inactivated FBS and Pen-Strep. HeLa S3 cells were obtained from Dr. Yoshihiro Nakatani's lab (Dana-Farber Cancer Institute) and cultured in DMEM with 5% FBS and Pen-Strep. Cells were verified by morphology match and known grow properties. Cells used did not show signs of mycoplasma contamination and were not tested for mycoplasma. U2OS cells were periodically treated with Plasmocin (Invivogen) to prevent mycoplasma contamination.

**Plasmids and mutagenesis**. The MCM10 cDNA was amplified by RT-PCR from RNA prepared from HeLa S3 cells and cloned between XhoI and NotI sites of the pOZ-FH-C[44] retroviral vector and a modified pEGFP-N1 vector (Clontech) with an added Myc tag at the N-terminus. pMG-BRCA2-GFP was previously described[45]. Site-directed mutagenesis was conducted according to the Quik-Change protocol (Agilent Technologies) to generate the MCM10 mutants used. Details will be provided upon request. Primers used for cloning, mutagenesis, and sequencing are provided in Supplementary Table 1. cDNA vectors encoding WT, CH (primase dead, polymerase competent), and AxA (catalytic dead) PRIMPOL proteins were described before[27] and kindly provided by Dr. Juan Méndez (The Spanish Cancer Research Centre).

**Identification of PALB2/BRCA2 binding proteins**. HeLa S3 cell line stably expressing FLAG-HA-double-tagged PALB2 and tandem purification of PALB2 complex were previously described[46]. Further details are provided in Supplementary Methods. The final purified materials were precipitated with acetone, dissolved and denatured in lithium dodecyl sodium (LDS) sample buffer and loaded onto a 4–12% Tris-Glycine SDS gel. The gel was run for ~1.5 cm and the entire sample-containing section of the gel lane (between the loading well and front dye) was excised and subjected to trypsin digestion and liquid chromatography (LC)-MS/MS analysis, which were conducted by the Biological Mass Spectrometry Facility of the Robert Wood Johnson Medical School and Rutgers, The State University of New Jersey.

**Generation of U2OS cell lines stably expressing MCM10 proteins**. Stable cell lines were generated by transducing cells with pOZ-FH-C-MCM10 retroviruses, and infected cells were selected using paramagnetic Dynabeads Goat Anti-Mouse IgG (Invitrogen, 11033) coupled with an antibody against IL2Rα (Millipore, 05-170). See Supplementary Methods for details.

**RNA interference**. siRNAs transfections were performed using Lipofectamine RNAiMax (ThermoFisher, 13778150) following manufacturer's instructions. The final concentration of siRNAs was 10 nmol/l. For western blotting, U2OS cells were plated at $1.5 \times 10^5$ cells per well in 6-well plates and harvested 48 h after siRNA transfection. For DNA fiber assay, cells were seeded in 12-well plates at $5 \times 10^4$ cells per well and labeled 48 h after transfection. Control siRNA AllStars was purchased from QIAGEN (1027281). All other siRNAs were custom-synthesized by Sigma Genosys. The sense strand sequences of the siRNAs are listed in Supplementary Table 2.

**DNA transfection and immunoprecipitation (IP)**. DNA transfections were performed using X-tremeGENE HP (Roche, 6366230001) following manufacturer's instructions. To analyze the association between various transiently expressed, FLAG-HA-tagged MCM10 proteins and endogenous BRCA2, 293T cells were seeded in 6-well plates at $5 \times 10^5$ cells per well and transfected with 1 μg of plasmid DNA and 3 μl of X-tremeGENE HP per well. Cells were harvested approximately 40 h after transfection and lysed in 350 μl of NETNG250 (20 mM Tris-HCl (pH 7.4)), 250 mM NaCl, 1 mM EDTA, 10 mM NaF, 0.5% NP-40, and 5% glycerol) supplemented with cOmplete® protease inhibitor cocktail (Roche, 11697498001). The tagged proteins were IPed from 300 μl of lysates with 5 μl of the Anti-FLAG Affinity Gel (Sigma, A2220) or Anti-HA-Agarose (Sigma, A2095). To test the association between MCM10 and BRCA2 variants, 293T cells were seeded as above and co-transfected with 1 μg of pOZC-MCM10 and 1 μg of pMG-BRCA2-GFP plasmids using 4.5 μl of X-tremeGENE HP. Cells were lysed as above and the BRCA2 variants were IPed using 1 μl of anti-GFP antibody (Thermo Fisher, A-11120) coupled with 5 μl of protein A agarose beads (Roche, 11134515001).

To detect the association between FLAG-HA-tagged MCM10 proteins with endogenous BRCA2 in the U2OS stable cells, cells were seeded into 6-well plates at $1.5 \times 10^5$ cells per well and allowed to adapt for 48 h. Cells were lysed in 300 μl of NETNG250, and IP was carried out using anti-FLAG M2 beads as above. To analyze the association between endogenous BRCA2 and MCM10, U2OS cells were seeded into 6-well plates at $1.5 \times 10^5$ cells per well and allowed to adapt for at least 48 h. Cells were lysed in 300 μl of NETNG250, and IP was carried out by adding 1 μl of anti-BRCA2 (Millipore Sigma, OP95) and 10 μl (slurry) of Protein G Agarose beads (Thermo Fisher, 15920010) to 200 μl of each lysate. Where applicable, cells were synchronized by a single thymidine block for 24 h and released for 2 h to enrich S-phase cells. All IPs were conducted by rocking the mixture at 4 °C overnight.

**Western blotting**. Cell lysates (12 μg per lane) or IP materials were heated in LDS sample buffer (Thermo Fisher, NP0007) at 72 °C for 15 min and electrophoresed on 4–12% gradient SDS-polyacrylamide gels. Proteins were transferred onto nitrocellulose membranes overnight at 4 °C. Blots were incubated with primary antibodies overnight at 4 °C and secondary antibodies for 1.5 h at RT and developed with Immobilon Western Chemiluminescent HRP Substrate (Millipore Sigma, WBKLS0500). The primary antibodies used are as follows: mouse anti-BRCA2 (Millipore Sigma, OP95, 1:2000 dilution), rabbit anti-MCM10 (Proteintech, 12251-1-AP, 1:1000), mouse anti-β-Actin (Santa Cruz, sc-69879, 1:10,000), rabbit anti-SMARCAL1 (Bethyl, A301-616A, 1:2000), rabbit anti-HLTF (Bethyl, A300-229A, 1:2000), rabbit anti-ZRANB3 (Proteintech, 23111-1-AP, 1:2000), rabbit anti-BRCA1 (Millipore, 07-434, 1:6000), mouse anti-GAPDH

(Proteintech, 60004-1-Ig, 1:100,000), and rabbit anti-RAD51 (Abcam, ab63801, 1:8000). Affinity purified rabbit anti-PALB2 (aa601-880) was described before[31] and used at 1:2000. Rat anti-PRIMPIOL was a gift from Dr. Juan Mendez (Spanish National Cancer Research Centre) and used at 1:200. Secondary antibodies used were Amersham ECL HRP-conjugated sheep anti-mouse IgG (GE Life Sciences, NA931-1ML, 1:8000), donkey anti-rabbit IgG (GE Life Sciences, NA934-1ML, 1:8000), and goat anti-rat antibody HRP conjugate (Sigma, AP183P, 1:8000).

**DNA fiber assay**. The procedures were performed largely following published protocols[20,47]. Briefly, active replication forks in cells were labeled by sequential 20 min pulses of two thymidine analogs, CldU (20 μM) followed by IdU (200 μM). Cells were then collected, washed, and resuspended in cold PBS at a density of $2 \times 10^3$ cells/μl. Two μl of the cell suspension were spotted to one end of the glass slide and lysed by 7.5 μl of lysis solution (50 mM EDTA and 0.5% SDS in 200 mM Tris-HCl (pH 7.4)) at room temperature (RT). DNA molecules were spread at a constant, low speed along the glass slides by tilting the slides by 15°. Slides were air-dried, fixed in 3:1 methanol/acetic acid at −20 °C for 15 min and stored at 4 °C overnight. Prior to staining, slides were washed with water, incubated in 2.5 M HCl for 80 min (to denature DNA), washed with PBS 3 times, and blocked with blocking buffer (5% BSA in PBS) for 20 min. Slides were then incubated at 37 °C for 2 h with primary antibodies (CldU: 1:300, rat monoclonal anti-BrdU, Abcam 6326; IdU: 1:30, mouse monoclonal anti-BrdU, BD biosciences 347580) diluted in blocking buffer. After three washes with PBS, slides were incubated with secondary antibodies (CldU: goat anti-rat, Alexa Fluor 594 (Invitrogen, A-11007), 1:300; IdU: goat anti-mouse, Alexa Fluor 488 (Invitrogen, A-11029), 1:300) in blocking buffer at 37 °C for 1 h. Slides were washed with PBS, air-dried, and were mounted with liquid mountant (ProLong Gold Antifade Mountant, Thermo Fisher, P36930). Images were acquired with a Nikon TE2000 fluorescence microscope with NIS Elements 4.40 software. For each sample, at least 10 images were taken from the whole slide and at least 200 individual IdU-labeled tracts were measured using ImageJ 1.58j1 software.

S1 nuclease digestion was conducted as previously described[28]. Immediately after thymidine analog pulse, cells were permeabilized with CSK100 buffer (100 mM NaCl, 10 mM MOPS, 3 mM MgCl₂, 300 mM sucrose, and 0.5% Triton X-100) for 10 min at RT, then incubated in S1 nuclease buffer (30 mM sodium acetate (pH 4.6), 10 mM zinc acetate, 5% glycerol, and 50 mM NaCl) with or without 20 U/ml S1 nuclease (Thermo Fisher, EN0321) for 15 min at 37 °C. Cells were then scraped in PBS + 0.1% BSA and centrifuged at 1500g at 4 °C. Cell pellets were resuspended in cold PBS at $2 \times 10^3$ cells/μl and then spotted and lysed on the slides. Antibody staining was performed as above.

**DNA combing assay**. DNA combing assay was carried out on the molecular combing system from Genomic Vision as described[48]. Cells were pulse labeled sequentially with 50 μM CldU and 50 μM IdU (20 min each) and then incubated with 500 μM thymidine for 60 min. Cells were harvested and embedded in 0.75% low melting agarose plugs. Agarose plugs were melted at 70 °C for 20 min in 0.1 M MES (2-(N-morpholino) ethanesulfonic acid) (pH 6.5). The melted agarose was cooled down to 42 °C and incubated in the presence of β Agarase I (NEB, M0392) overnight. Silanized coverslips were dipped in the DNA solution and incubated for 2 min at RT and then combed and baked for 2 h at 60 °C to crosslink DNA to coverslips. Next, slides were denatured in 0.4 M NaOH for 20 min at RT, washed extensively with PBS and then stained using CldU and IdU antibodies, and single DNA molecules were stained by YOYO-1.

**Chromosomal abnormality**. U2OS cells were seeded at 150,000 cells per well in 6-well plates and transfected with control (AllStars), BRCA2 (1949), PRIMPOL (pool of 1138 and 1158), or BRCA2 + PRIMPOL (BRCA2-1949 + pool of PRIMPOL-1138 and 1158) in 6-well plates. At 48 h after transfection, cells were subjected to 2 Gy of IR, and 6 h after IR, colcemid (0.2 μM) was added to enrich mitotic cells. Cells were harvested 90 min after colcemid addition, and chromosomal abnormality was measured by mitotic spread analysis as previously described[30]. Further details are provided in Supplementary Methods.

**Radiation and drug sensitivity**. U2OS cells were transfected with siRNAs as above. At 30 h after transfection, cells were trypsinized and reseeded into 96-well plates at 2000 cells per well and allowed to attach overnight. Cells were then treated with IR or drugs at different dosages. At 72 h after IR or drug addition, cell viability was determined using CellTiter-Glo® 2.0 Cell Viability Assay (G9241, Promega) according to the protocol provided by the manufacturer.

**Statistical analyses**. Statistical significance was determined by two-tailed unpaired Student's t test using GraphPad Prism 9.2.0.

## Data availability
The data supporting the findings of this study are available from the corresponding authors upon reasonable request. Uncropped western blot images are provided in the Source Data file.

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

## Acknowledgements

We thank Dr. Juan Mendez (Spanish National Cancer Research Centre) for provision of the PRIMPOL antibody and cDNA constructs, Dr. Maria Jasin (Memorial Sloan Kettering Cancer Center) for providing VC8 and VC8 + BRCA2 cells, and Dr. Inder Verma (Salk Institute) for providing the pMG-BRCA2-GFP plasmid. This work was supported by the US National Cancer Institute (R01CA138804, R01CA262227, and P01CA250957 to Z.S. and B.X.), the Robert Wood Johnson Foundation (through a grant to Rutgers Cancer Institute of New Jersey). Flow cytometry data were generated using the Rutgers Cancer Institute of New Jersey Flow Cytometry and Cell Sorting Shared Resource, which is supported, in part, by NCI-CCSG P30CA072720-5921. Z.K. and T.K.F were supported by postdoctoral fellowships from the New Jersey Commission on Cancer Research (NJCCR).

## Author contributions

B.X. conceived the project and supervised the work. B.X., Z.K., M.I.A, Z.S., L.Z., A.-K.B., and A.M. designed the experiments and interpreted the results; Z.K. performed DNA fiber, siRNA knockdown, IP, western blotting, and mitotic spread assays; P.F. performed DNA fiber, siRNA knockdown, and drug sensitivity assays; A.L.A. performed PALB2 complex purification, MCM10 cloning, mutagenesis, and MCM10–BRCA2 interaction domain mapping; H.F and C.R. performed DNA combing analyses; C.Y., T.K.F. and Y.Z. performed additional interaction domain mapping, mutagenesis and IP analyses; R. Baxley performed cell viability analysis; R. Buisson performed initial DNA fiber analysis; Z.K. and B.X. wrote the manuscript.

## Competing interests

The authors declare no competing interests.
