## [Peer Review File · Nature Communications]

BRCA2 associates with MCM10 to suppress PrimPol-mediated repriming and single-stranded DNA gap formation after DNA damageREVIEWER COMMENTS

Reviewer #1 (Remarks to the Author):

In this manuscript, the authors uncover a previously unappreciated function of BRCA2 in restraining PRIMPOL-repriming and ssDNA gap accumulation after DNA damage induction. Using single-molecule DNA fiber approaches, the authors show that BRCA2 restrains replication fork progression following treatment with ionizing radiation or bleomycin. Next, they show that unrestrained fork progression is due to the repriming activity of PRIMPOL which leads to longer tracts containing ssDNA gaps. Using a tandem affinity purification approach coupled to mass spectrometry, they find that BRCA2 interacts with MCM10. They further characterize this interaction *in vitro* and identify the coiled-coil motif at the N-terminus of MCM10 as essential for BRCA2 binding. Finally, they show that the BRCA2-MCM10 interaction is indeed important to restrain fork progression and prevent PRIMPOL-dependent repriming after DNA damage. Overall, this is an interesting story that points to a novel function of BRCA2, and its interaction with MCM10, in restraining PRIMPOL repriming. However, there are some major concerns that the authors should address to support their conclusions and strengthen their model.

Major criticisms:

1. The authors should test whether Rad51 depletion phenocopies the effect on fork progression observed upon BRCA2 depletion. This is particularly important as previous studies have shown that Rad51 depletion leads to PrimPol-dependent unrestrained replication progression or ssDNA gaps (Vallerga et al., PNAS, 2015). These previous findings should be adequately discussed in the manuscript.
2. In Figure 2, the authors knockdown factors implicated in replication fork reversal (SMARCAL1, HLF, and ZRANB3) and find that the loss of these fork reversal factors leads to increased fork velocity upon IR. Moreover, when they co-deplete BRCA2 with each of these factors they find a further increase in fork velocity. From these data, they conclude that unrestrained fork progression in BRCA2-deficient cells results from a mechanism which is different from the loss of fork reversal. To support their conclusions, the authors should demonstrate by co-depleting PRIMPOL that the increased fork velocity of SMARCAL1, HLF, and ZRANB3 deficient cells is indeed due to loss of fork reversal and not to PRIMPOL repriming.
3. The experiments of Figure 2D should include a control with BRCA2 wild-type cells to show that forks are slower in the presence of BRCA2 within the same graph.
4. Figure 2D. The authors should include a western to confirm that PRIMPOL has been efficiently depleted.
5. Figure 2F. The authors should include a control experiment with the S1 nuclease in cells co-depleted for BRCA2 and PRIMPOL to show that the S1-dependent shortening is no longer detected when PRIMPOL is depleted.
6. Figure 5. The authors should perform additional experiments to test their model that PRIMPOL recruitment is enhanced when the BRCA2-MCM10 interaction is compromised. For example, they could perform a chromatin fractionation or iPOND experiment to test whether there is increased recruitment of PRIMPOL to chromatin or to stalled forks, respectively.
7. The authors should include additional experiments to test how loss of PRIMPOL affects cell viability and genome stability in BRCA2-deficient cells after DNA damage induction and test how their newly discovered mechanism relates the radio-resistant DNA synthesis phenotype of BRCA2-deficient cells.
8. How do the authors explain their finding that the effect of IR or bleomycin on fork progression

increases with increased time of recovery?

9. The authors should test whether the function of BRCA2 in restraining fork progression is limited to conditions that induce DSBs (IR or bleomycin), or is a more general phenomenon also observed when cells are treated with other replication challenging agents that don't necessarily induce DSB accumulation, such as hydroxyurea or aphidicolin.

10. Extended data Figure 2b. How do the authors explain the fact that the D1-mutant of MCM10 lacking several residues at the N-terminus interacts with BRCA2 given that the coiled-coil motif located within the same domain of MCM10 is required for BRCA2 binding?

11. How do the authors reconcile their findings that loss of the MCM10-BRCA2 interaction increases PRIMPOL repriming with the recent study (also cited by the authors) that loss of MCM10 increases fork reversal (Mayle et al., PNAS 2019)?

Minor comments:

1. Scale bars should be added to the images of Figures 1b and 4h.

2. The authors should indicate in the figure legends whether fork progression is calculated by measuring the total track length (IdU+CldU) or the length of a single track (IdU or CldU).

3. The authors should indicate for how long they treat the cells with cisplatin, CPT, and bleomycin in the IP experiments of Figure 3E.

Reviewer #2 (Remarks to the Author):

In this study the authors examined the influence of BRCA2, PRIMPOL and MCM10 on replication fork movement and stability. They report that the primase/polymerase, PRIMPOL, is responsible for the radioresistant DNA synthesis observed in BRCA2-deficient cells. Moreover, they show that BRCA2 interacts with MCM10, and this interaction is important for the effectiveness of BRCA2 in repressing PRIMPOL activity. The authors invoke a model where PRIMPOL is involved in re-priming of DNA synthesis after DNA lesions, and BRCA2 suppresses this activity.

The topic of this study is interesting for a wide readership, as it concerns the important question of how damaged DNA is replicated, and how cell prevent catastrophic events during this process. The authors' experiments appear solid and are well described. However, I found that some of their most important conclusions are not fully supported by their data, as alternative interpretations have not been excluded. Thus, as it stands, I found the study a bit preliminary and in need of significant additional experimentation in order to become publishable in Nat. Comm. My major concerns are as follows:

1. At the outset of their study, the authors describe the radiation-resistant DNA synthesis (RDS) phenotype of BRCA2-deficient cells. They attribute that to an intra-S phase checkpoint defect; however, they do not address if this is indeed the case. Lack of intra-S checkpoint signaling has been suggested to control origin firing and fork speed. In that case, data from Lopes and colleagues suggest that ATR mediates a global fork slow-down via fork reversal (Mutreja et al. 2018 Cell Rep). Here the authors favor a model in which BRCA2 mediates fork slowing through a mechanism different from fork reversal. Hence, it is important to examine if the intra-S phase checkpoint is affected in BRCA2-deficient cells after IR.

2. Using almost exclusively IR (and to some extent HU) to study replication fork stalling and re-priming appears inappropriate, given that the most critical lesion induced by IR is a double-strand

break, which will have catastrophic effects on fork movement and cannot be overcome by re-priming. The cause of general replication stalling upon IR remains elusive. Given the dose of IR used, it is unlikely that all forks will run into DSBs to cause global fork slowing. The authors suggested in their model that the source of replisome stalling stems from base damage, where BRCA2 recruitment via MCM10 prevents repriming by PRIMPOL. If that is the case, one would expect a similar regulation when base lesions induced by other agents are present during replication. Recent studies have suggested a role of fork reversal as means to slow replication in the presence of wide range of DNA damaging agents (Zellweger et al. 2015 JCB). Therefore, in order to formally differentiate the MCM10-BRCA2 mechanism in fork slowing versus fork reversal, it is imperative that the authors test if replication speed-up can be observed upon other types of DNA damaging agents, such as UV, MMS, CPT and MMC, in BRCA2 deficient cells. Conceptually, a "lesion skipping" mechanism via PRIMPOL should only be possible for lesions that block replicative polymerases, e.g. UV photoproducts or MMS damage, but not for the ones that would hinder the helicase, e.g. MMC or CPT.

3. Following up on the previous point, the authors' model doesn't explain how lack of replication slowing occurs after bleomycin treatment in BRCA2-deficient cells (Figure 1h and Extended data Figure 1b). How would PRIMPOL-mediated repriming allow replication progression through a DSB? The authors should provide evidence that PRIMPOL is indeed responsible for the lack of fork slowing after bleomycin. Overall, these data seem to argue that the phenotype is rather due to a potential lack of checkpoint activation in BRCA2 mutant. In order to distinguish these scenarios, I would recommend that the authors perform DNA fiber analyses with a different labelling strategy, i.e. by first labeling ongoing forks with CIdU, followed by an IdU label in the presence of bleomycin. Changes of replication speed for ongoing forks can then be scored by measuring the lengths of IdU tracts in CIdU/IdU positive tracts, and new origin firing by counting the percentage of IdU-only tracts. A lack of suppression of new origin firing would be an indication of a lack of checkpoint activation.

4. The role of MCM10 in the process is underdeveloped. Although a physical interaction between BRCA2 and MCM10 is likely judging from the experiments shown here, this interaction has not been verified with recombinant proteins and could therefore still be indirect. This is important to exclude, given the localization of both proteins within the PALB2 complex.

5. The authors need to clearly differentiate their study from that by Quinet et al. (Mol Cell 2019), which also shows PRIMPOL-mediated re-priming in BRCA-deficient cells – but here replication fork reversal is invoked as an alternative mechanism.

Minor points

6. Analysis of DNA fiber data needs to be described more clearly. Did the authors only count the CIdU/IdU tracks to score the speed of ongoing forks? Did they observe any significant changes in new origin firing that would be indicative of lack of checkpoint control?

7. Figure 1. Is the role of BRCA2 in suppressing fork progression after IR related to its HR function or its fork protection function? Does a C-terminal truncation of BRCA2, which is defective in fork protection, give the same phenotype (Schlachter et al. 2011 Cell)? Alternatively, does perturbation of other HR factors such as BRCA1, Rad51, etc give rise to the same phenotype?

8. Figure 2f. According to the authors' model, PRIMPOL knockdown should prevent track shortening in the S1 nuclease assay.

9. Figure 3b. The level of transfected MCM10-FH seems to be very low or undetectable in the input. What is the reason for that?

10. Figure 4a-c and Extended data Fig 2. How do the authors explain their result where truncation of a larger domain in Mcm10 (Extended data Fig 2) did not abrogate Mcm10-Brc2 interaction but the

point mutants is able to do so (Figure 4a-c)?

11. The authors identified that BRCA2 interacts with MCM10 through its CC motif. Would they predict that BRCA2-MCM10 interaction will disrupt MCM10 self-association (Du et al. 2013 PLOS One)?

Reviewer #3 (Remarks to the Author):

This study by Kang et al investigates a mechanism by which BRCA2 suppresses replication fork elongation in presence of DNA damage. The {aberrant?} fork elongation observed in absence of BRCA2 depends on the primase PrimPol.

This study is timely, as there is growing interest in how PrimPol is regulated at replication forks encountering DNA damage, and how this interplays with homologous recombination factors such as BRCA1 (see Quinet et al., Mol Cell 2020). The manuscript is well-written, logical, and the data are mostly good quality (see caveats on IP below). The reported connection between BRCA2, MCM10 and PrimPol is novel. findings have the potential to influence thinking in the field.

However, this reviewer is not convinced that the results show a new mechanism of PrimPol-mediated fork elongation that is different from the already described by Quinet et al. That recent study reports that PrimPol-mediated repriming competes with replication fork regression and that inhibition of fork regression by depleting factors such as RAD51 increases PrimPol repriming activity. Like RAD51, BRCA2 is required for fork reversal and this could be the explanation for the observed phenotypes. The additive effect of the siRNAs in Figure 2D could be due to incomplete depletion of the individual proteins, leading to slightly better inhibition of fork regression when two different siRNAs are combined. The reported role for MCM10 could be related fork regression as well. The difference between the requirement for wild type MCM10 under the conditions e.g. in Figure 4h (IR and fork slowing) and Figure 4i (HU and fork degradation) could be due to different MCM10 requirements in response to these two DNA damaging agents.

Overall this study is very promising, but the proposed mechanism would need more thorough investigation. Overall the findings don't seem to go far enough to really support a new or different mechanism from what has been reported. One way forward might be to better explain the significance of the findings for the cell response to IR/bleomycin treatments, as the role of PrimPol in response to these is not well understood.

Queries:

Introduction and discussion are very short and should be extended, giving more background and discussion especially on PrimPol, MCM10 and ionizing radiation. Questions to expand on include:

- IR and bleomycin are used as DNA damaging agents. While there is no doubt that these treatments induce important replication-blocking lesions (base damage etc), the data are also harder to interpret as we don't know what specific types of lesions are causing the observed phenotypes. Why were IR and bleomycin chosen for this study? Bleomycin induces single-strand breaks and double-strand breaks, do the authors think that the observed effects occur in response to single-strand breaks?

- 10 Gy IR is a very high (cytotoxic) dose, what is the reasoning behind using it?

- Please discuss previous data on roles of PrimPol in repriming at lesions induced by IR or bleomycin. E.g., PrimPol-deficient cells were reported not to be hypersensitive to IR (Bianchi et al, Mol Cell 2013; Kobayashi et al., Cell Cycle 2016).

- The effect of 10 Gy IR on slowing fork progression is quite strong, especially early on after 1 or 3 hours release (Fig. 1D). The effect of BRCA2 depletion is small and forks are still significantly slowed in the absence of BRCA2 as well. This small effect is unlikely to explain RDS phenotypes observed in BRCA2-deficient backgrounds. The introduction, short as it is, mentions the RDS phenotype but how the present findings are thought to relate to it is not made clear.

Questions on data:

- The co-IP experiments to show the interaction between MCM10 and BRCA2 should have a DNase (e.g. bezonase) added to the extracts to exclude indirect interactions mediated by DNA. In vitro co-IPs of purified proteins are also helpful for this. Can a DNA-mediated interaction be excluded to make the results are more convincing?
- Why are MCM proteins shown in the Western blot in Fig. 1D?
- Why does D1 deletion mutant still interact with BRCA2 even though it lacks the coiled coil domain?
- What is the effect of MCM10 depletion on its own on fork speeds after IR? These data would really help interpret the overall results.

We thank all 3 reviewers for their critical and constructive reviews. The original critiques and our point-to-point responses are provided below.

Reviewer #1 (Remarks to the Author):

In this manuscript, the authors uncover a previously unappreciated function of BRCA2 in restraining PRIMPOL-repriming and ssDNA gap accumulation after DNA damage induction. Using single-molecule DNA fiber approaches, the authors show that BRCA2 restrains replication fork progression following treatment with ionizing radiation or bleomycin. Next, they show that unrestrained fork progression is due to the repriming activity of PRIMPOL which leads to longer tracts containing ssDNA gaps. Using a tandem affinity purification approach coupled to mass spectrometry, they find that BRCA2 interacts with MCM10. They further characterize this interaction in vitro and identify the coiled-coil motif at the N-terminus of MCM10 as essential for BRCA2 binding. Finally, they show that the BRCA2-MCM10 interaction is indeed important to restrain fork progression and prevent PRIMPOL-dependent repriming after DNA damage. Overall, this is an interesting story that points to a novel function of BRCA2, and its interaction with MCM10, in restraining PRIMPOL repriming. However, there are some major concerns that the authors should address to support their conclusions and strengthen their model.

Major criticisms:

1. The authors should test whether Rad51 depletion phenocopies the effect on fork progression observed upon BRCA2 depletion. This is particularly important as previous studies have shown that Rad51 depletion leads to PrimPol-dependent unrestrained replication progression or ssDNA gaps (Vallerga et al., PNAS, 2015). These previous findings should be adequately discussed in the manuscript.

Thanks for the suggestion. We have performed the DNA fiber assay in cells depleted of RAD51, alone or in combination with PRIMPOL, before and after IR (new Extended Data Figure 3). Indeed, we observed unrestrained fork progression in RAD51-depleted cells after IR; however, co-depletion of PRIMPOL with RAD51 only led to a very slight drop in replication tract length and failed to reverse the faster fork progression. This indicates that the unrestrained fork progression in RAD51-depleted cells after IR is not dependent on re-priming by PRIMPOL. The previous report has now been mentioned in the discussion of the revised manuscript.

2. In Figure 2, the authors knockdown factors implicated in replication fork reversal (SMARCAL1, HLTF, and ZRANB3) and find that the loss of these fork reversal factors leads to increased fork velocity upon IR. Moreover, when they co-deplete BRCA2 with each of these factors they find a further increase in fork velocity. From these data, they conclude that unrestrained fork progression in BRCA2-deficient cells results from a mechanism which is different from the loss of fork reversal. To support their conclusions, the authors should demonstrate by co-depleting PRIMPOL that the increased fork velocity of SMARCAL1, HLTF, and ZRANB3 deficient cells is indeed due to loss of fork reversal and not to PRIMPOL repriming.

We have performed the experiment as suggested (new Figure 2h). We found that co-depletion of PRIMPOL with SMARCAL1, HLTF, and ZRANB3 led to significant but modest reductions of fork progression after IR, compared with deletion of the any of the 3 fork remodeling factors alone. In comparison, co-depletion of PRIMPOL with BRCA2 led to more pronounced reduction in fork progression. These findings suggest that PrimPol-mediated repriming indeed operates in cells that have lost the 3 fork remodeling factors and

that fork reversal may preclude repriming by PrimPol. Moreover, the results also indicate that the increased fork velocity in cells depleted of the 3 factors is due to both reduced fork reversal and increased repriming by PrimPol.

3. The experiments of Figure 2D should include a control with BRCA2 wild-type cells to show that forks are slower in the presence of BRCA2 within the same graph.

Result of cells treated with a control siRNA has now been included. The forks are indeed slower in the presence of BRCA2.

4. Figure 2D. The authors should include a western to confirm that PRIMPOL has been efficiently depleted. PRIMPOL western has been added as new Figure 2e. The antibody was kindly provided by Dr. Juan Mendez at the Spanish National Cancer Research Centre.

5. Figure 2F. The authors should include a control experiment with the S1 nuclease in cells co-depleted for BRCA2 and PRIMPOL to show that the S1-dependent shortening is no longer detected when PRIMPOL is depleted.

We conducted the suggested experiment and have now added the results in the Figure (Fig. 2g now). Indeed, S1 nuclease treatment did not cause any further reduction in replication track length in BRCA2/PRIMPOL co-depleted cells, indicating that practically all ssDNA gaps in newly replicated DNA in the cells were generated by PRIMPOL-mediated repriming. Note that the PRIMPOL depletion was very efficient (Fig. 2e).

6. Figure 5. The authors should perform additional experiments to test their model that PRIMPOL recruitment is enhanced when the BRCA2-MCM10 interaction is compromised. For example, they could perform a chromatin fractionation or iPOND experiment to test whether there is increased recruitment of PRIMPOL to chromatin or to stalled forks, respectively.

We have analyzed PRIMPOL distribution in the chromatin fractions of cells expressing MCM10 Δ CC and 2A mutants that are defective in BRCA2 binding. Compared with cell expressing wt MCM10, cells expressing the two mutants both showed a slight reduction of PRIMPOL in the chromatin in unirradiated cells and a slight increase in irradiated cells. The experiment was done for a total of 4 times and the differences, albeit small, was statistically significant when using two-tailed paired t test. The new data has been added as Extended Data Figure 2. It is likely that BRCA2 only regulates a portion of PRIMPOL chromatin association, so we think a small difference is still reasonable.

7. The authors should include additional experiments to test how loss of PRIMPOL affects cell viability and genome stability in BRCA2-deficient cells after DNA damage induction and test how their newly discovered mechanism relates the radio-resistant DNA synthesis phenotype of BRCA2-deficient cells.

We have made substantial efforts to address all the above suggestions. First, we conducted mitotic spread analysis to determine the impact of PRIMPOL loss on chromosomal stability (new Figure 3a-c). The results showed that loss of PRIMPOL alone led to increased abnormalities in both unperturbed and irradiated cells; interestingly, compared with cells depleted of BRCA2 alone, co-depletion of PRIMPOL with BRCA2 caused increased chromosomal aberrations in unperturbed cells but reduced aberrations in irradiated cells. These results indicate that PRIMPOL and BRCA2 likely operate separately to maintain chromosomal stability during unperturbed DNA replication and that PRIMPOL-mediated repriming after IR in the absence of BRCA2 leads to increased DNA breakage presumably at ssDNA gaps. Second, we determined the viability

of cells depleted of PRIMPOL and BRCA2, both alone and in combination, to IR and bleomycin, as well as MMS and CPT (new Figure 3d-g). We found that depletion of PRIMPOL alone had little to no effect on cellular sensitivities to these agents but partially rescued the sensitivity of BRCA2-depleted cells to IR and bleomycin, which is consistent with the break-causing role of ssDNA gaps noted above. Third, we have used BrdU incorporation to semi-quantitatively measure rates of DNA synthesis in BRCA2-depleted cells and cells selectively expressing the MCM10 Δ CC mutant protein (Extended Data Figure 4). Our results showed that similar to BRCA2-depleted cells, cells expressing MCM10 Δ CC also undergo significant radio-resistant DNA synthesis, supporting the importance of the BRCA2-MCM10 interaction in restraining DNA synthesis after IR.

8. How do the authors explain their finding that the effect of IR or bleomycin on fork progression increases with increased time of recovery?

There may be multiple layers of mechanism underlying this observation. First, with increased time of recovery, there will be fewer and fewer physical blocks of fork progression as more and more DNA lesions are repaired, which can lead to a general recovery of fork velocity. Second, after IR, sequential ATM and ATR activation and attenuation will likely exert significant impact on replication factors and fork progression, which may promote recovery of fork speed at later time points. After IR and bleomycin treatment, the difference between BRCA2 proficient and deficient cells becomes larger at later time points, which could be due to a number of reasons such as the higher overall fork speed, temporally regulated interplay between BRCA2 and ATM/ATR, or DDR-related regulations of MCM10 and PRIMPOL. These possibilities remain to be investigated.

9. The authors should test whether the function of BRCA2 in restraining fork progression is limited to conditions that induce DSBs (IR or bleomycin), or is a more general phenomenon also observed when cells are treated with other replication challenging agents that don't necessarily induce DSB accumulation, such as hydroxyurea or aphidicolin.

We have performed similar experiments with cells treated with hydroxyurea (HU), camptothecin (CPT) and methyl methanesulfonate (MMS). BRCA2-depleted cells showed faster fork progression than control cells after all 3 drug treatments, indicating that the role of BRCA2 in restraining fork progression after DNA damage is a general phenomenon.

10. Extended data Figure 2b. How do the authors explain the fact that the D1-mutant of MCM10 lacking several residues at the N-terminus interacts with BRCA2 given that the coiled-coil motif located within the same domain of MCM10 is required for BRCA2 binding?

We noticed that the expression levels of myc-MCM10-GFP proteins (expressed from a pcDNA3-based vector) were extremely high and they form large aggregates or droplet-like structures in the cell when viewed under a fluorescence microscope. Therefore, we strongly suspected that the co-IP of BRCA2 with MCM10 lacking its N terminus was an artifact. We then happened to turn our attention to the CC motif of MCM10, because we had been working on the interaction between the CC motifs of BRCA1 and PALB2 for several years. Once we found that deletion of the CC motif of MCM10 (in the pOZ-FH-C retroviral vector) abrogated its interaction with MCM10, we stopped further mapping using other approaches. Soon after we started to focus on the CC motif, its structure was published (Du *et al.* Plos One 2013), which led us to generate the 2A, 2D and 4A mutations in the CC motif.

11. How do the authors reconcile their findings that loss of the MCM10-BRCA2 interaction increases PRIMPOL repriming with the recent study (also cited by the authors) that loss of MCM10 increases fork reversal (Mayle et al., PNAS 2019)?

Our study is different from the Mayle study in that we were focusing on the effects of loss of the MCM10-BRCA2 interaction rather than loss of the MCM10 protein.

Minor comments:

1. Scale bars should be added to the images of Figures 1b and 4h.

Scale bars have been added.

2. The authors should indicate in the figure legends whether fork progression is calculated by measuring the total track length (IdU+CldU) or the length of a single track (IdU or CldU).

Fork progression was all calculated based on IdU track length, which in our experience was more accurate. This information has been added to the legends in Figure 1 and also specified in the methods section.

3. The authors should indicate for how long they treat the cells with cisplatin, CPT, and bleomycin in the IP experiments of Figure 3E.

Thanks for pointing this out. The treatments were for 3h. This has been added.

Reviewer #2 (Remarks to the Author):

In this study the authors examined the influence of BRCA2, PRIMPOL and MCM10 on replication fork movement and stability. They report that the primase/polymerase, PRIMPOL, is responsible for the radioresistant DNA synthesis observed in BRCA2-deficient cells. Moreover, they show that BRCA2 interacts with MCM10, and this interaction is important for the effectiveness of BRCA2 in repressing PRIMPOL activity. The authors invoke a model where PRIMPOL is involved in re-priming of DNA synthesis after DNA lesions, and BRCA2 suppresses this activity.

The topic of this study is interesting for a wide readership, as it concerns the important question of how damaged DNA is replicated, and how cell prevent catastrophic events during this process. The authors' experiments appear solid and are well described. However, I found that some of their most important conclusions are not fully supported by their data, as alternative interpretations have not been excluded. Thus, as it stands, I found the study a bit preliminary and in need of significant additional experimentation in order to become publishable in Nat. Comm. My major concerns are as follows:

1. At the outset of their study, the authors describe the radiation-resistant DNA synthesis (RDS) phenotype of BRCA2-deficient cells. They attribute that to an intra-S phase checkpoint defect; however, they do not address if this is indeed the case. Lack of intra-S checkpoint signaling has been suggested to control origin firing and fork speed. In that case, data from Lopes and colleagues suggest that ATR mediates a global fork slow-down via fork reversal (Mutreja et al. 2018 Cell Rep). Here the authors favor a model in which BRCA2 mediates fork slowing through a mechanism different from fork reversal. Hence, it is important to examine if the intra-S phase checkpoint is affected in BRCA2-deficient cells after IR.

As stated in our response to reviewer #1, we have conducted semi-quantitative measurement of DNA synthesis using BrdU incorporation. As shown in the new Extended Data Figure 4, depletion of BRCA2 led to increased DNA synthesis, especially in late S phase cells, after IR, and a similar increase was seen in cells selectively expressing MCM10 Δ CC which is largely unable to bind BRCA2. Therefore, we think the MCM10-BRCA2 interaction does contribute to the intra-S phase checkpoint. However, this is by no means to say that what we found is the sole or major mechanism of the checkpoint control. We would also like to note that although this study was originally motivated in part by a desire to elucidate the mechanism of BRCA2 regulation of intra-S phase checkpoint, it is no longer the focus of the current manuscript.

2. Using almost exclusively IR (and to some extent HU) to study replication fork stalling and re-priming appears inappropriate, given that the most critical lesion induced by IR is a double-strand break, which will have catastrophic effects on fork movement and cannot be overcome by re-priming. The cause of general replication stalling upon IR remains elusive. Given the dose of IR used, it is unlikely that all forks will run into DSBs to cause global fork slowing. The authors suggested in their model that the source of replisome stalling stems from base damage, where BRCA2 recruitment via MCM10 prevents repriming by PRIMPOL. If that is the case, one would expect a similar regulation when base lesions induced by other agents are present during replication. Recent studies have suggested a role of fork reversal as means to slow replication in the presence of wide range of DNA damaging agents (Zellweger et al. 2015 JCB). Therefore, in order to formally differentiate the MCM10-BRCA2 mechanism in fork slowing versus fork reversal, it is imperative that the authors test if replication speed-up can be observed upon other types of DNA damaging agents, such as UV, MMS, CPT and MMC, in BRCA2 deficient cells. Conceptually, a “lesion skipping” mechanism via PRIMPOL should only be possible for lesions that block replicative polymerases, e.g. UV photoproducts or MMS damage, but not for the ones that would hinder the helicase, e.g. MMC or CPT.

As noted above, we mainly used IR because the study was originally intended to address the radio-resistant DNA synthesis (RDS) phenotype of BRCA2 deficient cells. We have now tested MMS, CPT and HU (new Figure 1g). We found that BRCA2-depleted cells and cells selectively expressing MCM10 Δ CC also showed unrestrained fork progression after these 3 drug treatments. Importantly, the difference was largest after MMS treatment. These results indicate that repriming across base damage may be the most frequent events catalyzed by PrimPol in the above cells after DNA damage. In case of CPT, PRIMPOL is likely to only “kick in” after the bulk of covalently linked TOP1 on DNA has been cleaved off by a protease such as SPRTN leaving only a small stub on DNA that may be bypassed by the MCM2-7 helicase complex.

3. Following up on the previous point, the authors’ model doesn’t explain how lack of replication slowing occurs after bleomycin treatment in BRCA2-deficient cells (Figure 1h and Extended data Figure 1b). How would PRIMPOL-mediated repriming allow replication progression through a DSB? The authors should provide evidence that PRIMPOL is indeed responsible for the lack of fork slowing after bleomycin. Overall, these data seem to argue that the phenotype is rather due to a potential lack of checkpoint activation in BRCA2 mutant. In order to distinguish these scenarios, I would recommend that the authors perform DNA fiber analyses with a different labelling strategy, i.e. by first labeling ongoing forks with CIdU, followed by an IdU label in the presence of bleomycin. Changes of replication speed for ongoing forks can then be scored by measuring the lengths of IdU tracts in CIdU/IdU positive tracts, and new origin firing by counting the percentage of IdU-only tracts. A lack of suppression of new origin firing would be an indication of a lack of checkpoint activation.

Similar to ionizing radiation, bleomycin causes DNA damage through generation of reactive oxygen species,

especially hydroxyl radicals and superoxide, which in fact may cause more base damage than DNA breaks. Our model by no means suggests that PRIMPOL can bypass a DSB. We have also performed the recommended experiment (new Extended Data Figure 5). In brief, we first labeled replicating DNA in cells with CldU for 20 min, then labeled them with IdU in the presence of bleomycin for 40 min, and then analyzed replication tracts without and with S1 nuclease treatment. Similar to the results after IR, cells depleted of BRCA2 showed increased replication tract length along with more ssDNA gaps, and combined depletion of BRCA2 and PRIMPOL reversed the unrestrained fork progression and eliminated the gap formation. We also quantified new origin firing by counting IdU-only tracts as suggested, and we found that overall origin firing was reduced in the presence of bleomycin and that cells depleted of BRCA2 showed even lower (though not statistically significant) origin firing compared with control cells, indicating that checkpoint activation is at least normal in terms of origin suppression.

4. The role of MCM10 in the process is underdeveloped. Although a physical interaction between BRCA2 and MCM10 is likely judging from the experiments shown here, this interaction has not been verified with recombinant proteins and could therefore still be indirect. This is important to exclude, given the localization of both proteins within the PALB2 complex.

To demonstrate the interaction is direct, we would need to produce recombinant BRCA2, either full-length or a fragment that is responsible for MCM10 binding. Given the very large size of BRCA2 (384KD), purifying full-length BRCA2 free from its binding proteins for in vitro assay is beyond the expertise of our lab (and most labs). We therefore attempted to map the binding site of MCM10 on BRCA2 in order to identify a fragment that is responsible for MCM10 binding. Our results appeared to suggest that there are at least 2 MCM10 binding sites in BRCA2, but regrettably, we were unable to obtain any clear results in our in vitro binding assay. Therefore, we decided to continue to work on this problem in the future. Still, we were able to rule out the possibility that the interaction observed may be mediated by the trace amount of chromatin DNA in the cell lysate by treating the final IPed material (on anti-FLAG beads) with DNase I (and then wash again) prior to running them on the gel for western blotting. Given our inability to demonstrate a direct interaction, we have changed the phrase “BRCA2 interacts with MCM10” in the title to “BRCA2 associates with MCM10” and similarly changed the word “interaction” to “association” in the text. We have also modified our model by adding a protein “X” in the complex, which might be needed to bridge the BRCA2-MCM10 association if the interaction is indirect.

5. The authors need to clearly differentiate their study from that by Quinet et al. (Mol Cell 2019), which also shows PRIMPOL-mediated re-priming in BRCA-deficient cells – but here replication fork reversal is invoked as an alternative mechanism.

The Quinet study emphasizes a role for (induced and accumulated) PRIMPOL to reprime DNA synthesis beyond cisplatin-induced crosslinks in BRCA1-deficient cells, thereby precluding fork reversal and reducing degradation of reversed forks. In our study, we present a BRCA2-specific mechanism that is also independent of fork reversal. This has been added in the discussion.

Minor points

6. Analysis of DNA fiber data needs to be described more clearly. Did the authors only count the CldU/IdU tracks to score the speed of ongoing forks? Did they observe any significant changes in new origin firing that would be indicative of lack of checkpoint control?

We did score origins as well but we did not see any evidence that would be indicative of lack of checkpoint control in terms of origin regulation. In fact, as described above, BRCA2-depleted cells showed slightly reduced, rather than increased, new origin firing in the presence of bleomycin.

7. Figure 1. Is the role of BRCA2 in suppressing fork progression after IR related to its HR function or its fork protection function? Does a C-terminal truncation of BRCA2, which is defective in fork protection, give the same phenotype (Schlachter et al. 2011 Cell)? Alternatively, does perturbation of other HR factors such as BRCA1, Rad51, etc give rise to the same phenotype?

We have tested both BRCA1 and RAD51. We did not see any similarity between BRCA1 and BRCA2 in any aspects described in the manuscript. Depletion of RAD51 indeed led to faster fork progression after DNA damage; however, unlike BRCA2, the increased fork velocity in RAD51-depleted cells was not due to PRIMPOL mediated repriming, as co-depletion of PRIMPOL with RAD51 produced little effect on fork progression (new Extended Data Figure 3). These results indicate that the fork-restraining function of BRCA2 is independent of its function in HR.

8. Figure 2f. According to the authors' model, PRIMPOL knockdown should prevent track shortening in the S1 nuclease assay.

We have added data on siBRCA2+siPRIMPOL+S1 nuclease treatment (new Figure 2h). Indeed, PRIMPOL knockdown prevented shortening of replication tracks by S1 nuclease treatment, which lends further support of our model.

9. Figure 3b. The level of transfected MCM10-FH seems to be very low or undetectable in the input. What is the reason for that?

The sample in the INPUT lane was lysate from cells transfected with the empty vector (pOZ-FH-C) to give a sense of the endogenous levels of PALB2 and BRCA2, and the antibody used to detect the overexpressed MCM10 was anti-HA, which will not detect the endogenous MCM10. We have revised the labels to clarify these. In the new Figure 5b, both endogenous and transiently expressed MCM10 can be seen.

10. Figure 4a-c and Extended data Fig 2. How do the authors explain their result where truncation of a larger domain in Mcm10 (Extended data Fig 2) did not abrogate Mcm10-Brca2 interaction but the point mutants is able to do so (Figure 4a-c)?

As mentioned in our answer to Reviewer #1, we noticed that the expression levels of myc-MCM10-GFP proteins (expressed from a pcDNA3-based vector) were extremely high and they formed large aggregates in the cell when viewed under a fluorescence microscope. Therefore, we believe that the co-IP of BRCA2 with MCM10 lacking its N terminus was an artifact. We then turned our attention to the CC motif of MCM10, because we had been working on the interaction between the CC motifs of BRCA1 and PALB2 for several years. Indeed, deletion of the CC motif of MCM10 (in a retroviral vector) abrogated its association with BRCA2. After we started to focus on the CC motif, its structure was published (Du *et al.* Plos One 2013), which led us to further generate the 2A, 2D and 4A mutations in the motif.

11. The authors identified that BRCA2 interacts with MCM10 through its CC motif. Would they predict that BRCA2-MCM10 interaction will disrupt MCM10 self-association (Du et al. 2013 PLOS One)?

We think it is possible. However, to us it is unlikely that BRCA2 binding to MCM10 would disrupt a major fraction of MCM10 self-association to a point where it affects other functions of MCM10. Also, the

functional importance of MCM10 self-association is unclear.

Reviewer #3 (Remarks to the Author):

This study by Kang et al investigates a mechanism by which BRCA2 suppresses replication fork elongation in presence of DNA damage. The {aberrant(?) } fork elongation observed in absence of BRCA2 depends on the primase PrimPol.

This study is timely, as there is growing interest in how PrimPol is regulated at replication forks encountering DNA damage, and how this interplays with homologous recombination factors such as BRCA1 (see Quinet et al., Mol Cell 2020). The manuscript is well-written, logical, and the data are mostly good quality (see caveats on IP below). The reported connection between BRCA2, MCM10 and PrimPol is novel. findings have the potential to influence thinking in the field.

However, this reviewer is not convinced that the results show a new mechanism of PrimPol-mediated fork elongation that is different from the already described by Quinet et al. That recent study reports that PrimPol-mediated repriming competes with replication fork regression and that inhibition of fork regression by depleting factors such as RAD51 increases PrimPol repriming activity. Like RAD51, BRCA2 is required for fork reversal and this could be the explanation for the observed phenotypes. The additive effect of the siRNAs in Figure 2D could be due to incomplete depletion of the individual proteins, leading to slightly better inhibition of fork regression when two different siRNAs are combined. The reported role for MCM10 could be related fork regression as well. The difference between the requirement for wild type MCM10 under the conditions e.g. in Figure 4h (IR and fork slowing) and Figure 4i (HU and fork degradation) could be due to different MCM10 requirements in response to these two DNA damaging agents.

Overall this study is very promising, but the proposed mechanism would need more thorough investigation. Overall the findings don't seem to go far enough to really support a new or different mechanism from what has been reported. One way forward might be to better explain the significance of the findings for the cell response to IR/bleomycin treatments, as the role of PrimPol in response to these is not well understood.

Queries:

Introduction and discussion are very short and should be extended, giving more background and discussion especially on PrimPol, MCM10 and ionizing radiation. Questions to expand on include:

- IR and bleomycin are used as DNA damaging agents. While there is no doubt that these treatments induce important replication-blocking lesions (base damage etc.), the data are also harder to interpret as we don't know what specific types of lesions are causing the observed phenotypes. Why were IR and bleomycin chosen for this study? Bleomycin induces single-strand breaks and double-strand breaks, do the authors think that the observed effects occur in response to single-strand breaks?

First, we have now added data from cells treated with MMS, CPT and HU (new Figure 1g). Among the 3 drugs, MMS treatment appeared to reveal the largest difference between control and BRCA2-deficient cells, suggesting that base damage and/or the ensuing SSBs may be the most relevant lesion for the mechanism presented. Second, as mentioned in our response to Reviewer #1, we chose IR because the study was originally intended to address the mechanism of the previously reported RSD phenotype of BRCA2 deficient cells. Third, given the difference seen after HU treatment, we think the mechanism may also be operative in

response to ssDNA gaps.

- 10 Gy IR is a very high (cytotoxic) dose, what is the reasoning behind using it?

The use of 10 Gy dose had a historic reason in the lab. Although it is undoubtedly a high dose, U2OS cells tolerate it quite well and will progress through the S phase and eventually arrest in G2 with no cell death within 24h (longer time not tested). Since our longest time point in this study is 6h after radiation, we do not believe cytotoxicity will affect our results in any major way. For most experiments we also used 2 Gy, which is very close to a typical fraction of radiotherapy.

- Please discuss previous data on roles of PrimPol in repriming at lesions induced by IR or bleomycin. E.g., PrimPol-deficient cells were reported not to be hypersensitive to IR (Bianchi et al, Mol Cell 2013; Kobayashi et al., Cell Cycle 2016).

We have now directly tested the effect of PRIMPOL knockdown on cellular sensitivity to IR and bleomycin (new Figure 3e, f). Indeed, loss of PRIMPOL had little to no effect, consistent with the above previous reports. The above reports have also been cited.

- The effect of 10 Gy IR on slowing fork progression is quite strong, especially early on after 1 or 3 hours release (Fig. 1D). The effect of BRCA2 depletion is small and forks are still significantly slowed in the absence of BRCA2 as well. This small effect is unlikely to explain RDS phenotypes observed in BRCA2-deficient backgrounds. The introduction, short as it is, mentions the RDS phenotype but how the present findings are thought to relate to it is not made clear.

As stated in our response to reviewer #1, we have conducted semi-quantitative measurement of DNA synthesis using BrdU incorporation. As shown in the new Extended Data Figure 4, depletion of BRCA2 led to increased DNA synthesis, especially in late S phase cells, after IR, and a similar increase was seen in cells selectively expressing MCM10 Δ CC which is unable to bind BRCA2. Therefore, we think the MCM10-BRCA2 association does contribute to the intra-S phase checkpoint. However, this is no means to say that what we found is the sole or major mechanism of the checkpoint control. We would also like to note that although this study was originally motivated by a desire to elucidate the mechanism of the RDS phenotype, it is no longer the focus of the current manuscript.

Questions on data:

- The co-IP experiments to show the interaction between MCM10 and BRCA2 should have a DNase (e.g. bezonase) added to the extracts to exclude indirect interactions mediated by DNA. In vitro co-IPs of purified proteins are also helpful for this. Can a DNA-mediated interaction be excluded to make the results more convincing?

We have used two methods to rule out the possibility of DNA-mediated interaction. First, we used DNase to treat the cell lysate then performed IP. Second, we first performed IP and then used DNase I to treat the IPed material on beads. In both cases, the co-IP of MCM10 and BRCA2 was unaffected. New data which includes DNase I treatment of beads is now presented in Figure 4e. Note that in our hands treating beads post IP or pulldown is a very effective way of eliminating DNA-mediated associations.

- Why are MCM proteins shown in the Western blot in Fig. 1D?

They were tested in parallel with BRCA2 for curiosity. MCM4 and MCM6 have been removed, MCM10 is kept there because it is somewhat relevant.

- Why does D1 deletion mutant still interact with BRCA2 even though it lacks the coiled coil domain?

As mentioned in our answers to Reviewers #1 and #2, the expression levels of myc-MCM10-GFP proteins (expressed from a pcDNA3-based vector) were extremely high and the proteins formed large aggregates in the cell. Therefore, we believe that the co-IP of BRCA2 with MCM10 lacking its N terminus was an artifact. We then turned our attention to the CC motif of MCM10, because we had been working on the interaction between the coiled-coil motifs of BRCA1 and PALB2 for several years.

- What is the effect of MCM10 depletion on its own on fork speeds after IR? These data would really help interpret the overall results.

We have tested the effect of MCM10 depletion on fork speed along with BRCA2 depletion. Consistent with previous reports that MCM10 promotes fork progression, our results show that loss of MCM10 significantly slows down fork progression both before and after IR (new Extended Data Figure 4). The data clearly demonstrate that loss of MCM10-BRCA2 association has a fundamentally different impact on replication fork progression from loss of MCM10 protein.

REVIEWER COMMENTS

Reviewer #1 (Remarks to the Author):

The authors have done an excellent job in addressing the reviewers' concerns. The revised version of the manuscript is significantly improved but there are still some issues that remain to be addressed:

1. Figure 2. The authors should repeat the fiber assays of Figure 2 with a catalytically-dead PRIMPOL mutant to support their conclusion that the repriming activity of PRIMPOL promotes ssDNA gap accumulation and unrestrained fork progression.
2. Pg. 11, lines 234-236: The authors should include a western blot to confirm that MCM10 is efficiently depleted by the two siRNAs against the 3'-UTR.
3. Pg. 12, lines 270-271: The authors should include western blot to confirm that PRIMPOL is efficiently depleted in combination with MCM10.
4. Pg. 15, lines 325-327: The expression levels of BRCA2 are considerably reduced in the RAD51 depleted cells, as also noted by the authors in the legend of Extended Figure 3A. Have the authors considered the possibility that the unrestrained fork progression phenotype observed in the RAD51 depleted cells might be simply associated to the decreased expression of BRCA2?
5. Pg. 15, lines, 334-335: The authors state that the PRIMPOL-mediated pathway is specific to BRCA2-deficient cells because it is not observed in BRCA1-depleted cells. The data with the BRCA1-depleted cells should be included in the manuscript to support this conclusion.
6. Figure 2A. The expression levels of SMARCAL1 are considerably reduced in the ZRANB3 depleted cells. The authors should discuss the possible reasons for this effect.
7. Pg. 10, lines 214-216: "Extended Figure 2" should be corrected with "Extended Figure 1".

Reviewer #2 (Remarks to the Author):

The authors have performed a series of additional experiments and have made revisions to the text, which in combination has significantly strengthened their arguments. I am now happy with the manuscript as is.

Reviewer #3 (Remarks to the Author):

The authors have addressed the reviewer's concerns very well and the manuscript is much improved. In my view, the results of study are still fairly similar to previous reports from the Vindigni and Gottifredi labs. It is a shame that the data after BRCA1 depletion are not shown, as they would help to further differentiate the findings.

Minor comments:

New Figure 3 c, d: It would be beneficial to have a clearer legend in the graphs, ideally including the bar colors, as the symbols are too small to see.

Legend to extended data Figure 3: It should read RAD51, not RDA51.

Reviewer #1 (Remarks to the Author):

The authors have done an excellent job in addressing the reviewers' concerns. The revised version of the manuscript is significantly improved but there are still some issues that remain to be addressed:

1. Figure 2. The authors should repeat the fiber assays of Figure 2 with a catalytically-dead PRIMPOL mutant to support their conclusion that the repriming activity of PRIMPOL promotes ssDNA gap accumulation and unrestrained fork progression.

We have obtained the cDNA constructs encoding wt, the "AxA" catalytic dead mutant, and the "CH" primase dead mutant PRIMPOL from Dr. Juan Mendez at the Spanish National Cancer Research Centre. These constructs were transiently expressed in BRCA2/PRIMPOL doubly depleted cells and their ability to restore unrestrained fork progression was analyzed (new Extended Data Figure 1). Indeed, wt PRIMPOL was able to do so while neither mutant was. This data clearly demonstrates that PRIMPOL and its enzymatic activities underlie the unrestrained fork progression in BRCA2 deficient cells after DNA damage.

2. Pg. 11, lines 234-236: The authors should include a western blot to confirm that MCM10 is efficiently depleted by the two siRNAs against the 3'-UTR.

MCM10 depletion by the two 3'-UTR siRNAs is now shown in Extended Data Figure 2. In fact, we think that since the exogenous MCM10 proteins are overexpressed, depletion of the endogenous protein may not be necessary. Also, in light of the next question below, we conducted MCM10 knockdown along with co-depletion of MCM10 with PRIMPOL, which is shown in the same figure.

3. Pg. 12, lines 270-271: The authors should include western blot to confirm that PRIMPOL is efficiently depleted in combination with MCM10.

Please see above. Interestingly, we found that depletion of the endogenous MCM10 with the two 3'-UTR siRNAs led to reduced PRIMPOL amount and that depletion of PRIMPOL with the two siRNAs used caused increased expression of MCM10. While neither of these would affect the interpretation of the results, these are worth of further investigation nonetheless.

4. Pg. 15, lines 325-327: The expression levels of BRCA2 are considerably reduced in the RAD51 depleted cells, as also noted by the authors in the legend of Extended Figure 3A. Have the authors considered the possibility that the unrestrained fork progression phenotype observed in the RAD51 depleted cells might be simply associated to the decreased expression of BRCA2?

We think it is highly unlikely that the effect of RAD51 depletion was simply due to the moderately reduced BRCA2 amount. First, the reduction in BRCA2 is not very severe. Second, if

the unrestrained fork progression was indeed due to reduced BRCA2 amount, it should be reversed by PRIMPOL depletion, but that was clearly not the case. Therefore, we think RAD51 and BRCA2 restrain fork progression through distinct mechanisms.

5. Pg. 15, lines, 334-335: The authors state that the PRIMPOL-mediated pathway is specific to BRCA2-deficient cells because it is not observed in BRCA1-depleted cells. The data with the BRCA1-depleted cells should be included in the manuscript to support this conclusion.

We found that depletion of BRCA1 in U2OS cells leads to reduced fork speed even without exogenous DNA damage or replication stress, and fork speed in these cells further decreased and remained much slower than control after IR (new Extended Data Figure 5). Thus, BRCA1 and BRCA2 have opposite functions in this regard. This is a very interesting finding that we plan to continue to investigate.

6. Figure 2A. The expression levels of SMARCAL1 are considerably reduced in the ZRANB3 depleted cells. The authors should discuss the possible reasons for this effect.

Actually, our original data showed that ZRANB3 levels were considerably lower in SMARCAL1 depleted cells, rather than SMARCAL1 levels being lower in ZRANB3 depleted cells. We have repeated the knockdown and western experiment multiple times and found that ZRANB3 siRNAs indeed have no effect on SMARCAL1 expression, while one of the two SMARCAL1 siRNAs did cause reduction in ZRANB3 amount. The blots have now been replaced. While we can only speculate about the cause of ZRANB3 downregulation, our data at least show that the overall effect caused by SMARCAL1 siRNAs is not due to ZRANB3 loss.

7. Pg. 10, lines 214-216: “Extended Figure 2” should be corrected with “Extended Figure 1”.

Thanks for the catch. We have now added a new Extended Data Figure 1, therefore that Figure is indeed Extended Data Figure 2 now.

Reviewer #2 (Remarks to the Author):

The authors have performed a series of additional experiments and have made revisions to the text, which in combination has significantly strengthened their arguments. I am now happy with the manuscript as is.

Thanks for your support!

Reviewer #3 (Remarks to the Author):

The authors have addressed the reviewer's concerns very well and the manuscript is much

improved. In my view, the results of study are still fairly similar to previous reports from the Vindigni and Gottifredi labs. It is a shame that the data after BRCA1 depletion are not shown, as they would help to further differentiate the findings.

The data after BRCA1 knockdown are now shown in Extended Data Figure 5. As noted in the above response to the first reviewer, BRCA1 depletion led to completely different outcomes from BRCA2 depletion in the setting used in our study.

Minor comments:

New Figure 3 c, d: It would be beneficial to have a clearer legend in the graphs, ideally including the bar colors, as the symbols are too small to see.

The legends are now enlarged and bars made clearer by using more visible colors.

Legend to extended data Figure 3: It should read RAD51, not RDA51.

Corrected. Thanks.

REVIEWERS' COMMENTS

Reviewer #1 (Remarks to the Author):

The authors properly addressed all the remaining concerns of this reviewer. The manuscript is significantly strengthened and there are no further issues to be addressed.

The only minor comment is that PRIMPOL is misspelled in Extended Figure 3.

Reviewer #1 (Remarks to the Author):

The authors properly addressed all the remaining concerns of this reviewer. The manuscript is significantly strengthened and there are no further issues to be addressed.

The only minor comment is that PRIMPOL is misspelled in Extended Figure 3.

This has been corrected.